# High precision epidermal radio frequency antenna via nanofiber network for wireless stretchable multifunction electronics

Yufei Zhang [1,2,6], Zhihao Huo[1,2,6], Xiandi Wang [1,2,6✉], Xun Han[1,2,3], Wenqiang Wu[1], Bensong Wan[1], Hui Wang[4], Junyi Zhai [1,2], Juan Tao[1,2], Caofeng Pan [1,2,3✉] & Zhong Lin Wang [1,2,5✉]

Recently, stretchable electronics combined with wireless technology have been crucial for realizing efficient human-machine interaction. Here, we demonstrate highly stretchable transparent wireless electronics composed of Ag nanofibers coils and functional electronic components for power transfer and information communication. Inspired by natural systems, various patterned Ag nanofibers electrodes with a net structure are fabricated via using lithography and wet etching. The device design is optimized by analyzing the quality factor and radio frequency properties of the coil, considering the effects of strain. Particularly, the wireless transmission efficiency of a five-turn coil drops by approximately only 50% at 10 MHz with the strain of 100%. Moreover, various complex functional wireless electronics are developed using near-field communication and frequency modulation technology for applications in content recognition and long-distance transmission (>1 m), respectively. In summary, the proposed device has considerable potential for applications in artificial electronic skins, human healthcare monitoring and soft robotics.

---

[1] CAS Center for Excellence in Nanoscience, Beijing Key Laboratory of Micro-nano Energy and Sensor, Beijing Institute of Nanoenergy and Nanosystems, Chinese Academy of Sciences, 100083 Beijing, China. [2] School of Nanoscience and Technology, University of Chinese Academy of Sciences, 100049 Beijing, China. [3] College of Optoelectronic Engineering, Shenzhen University, 518060 Shenzhen, China. [4] Key Laboratory of Aerospace Materials and Performance (Ministry of Education), School of Materials Science and Engineering, Beihang University, 100191 Beijing, China. [5] School of Materials Science and Engineering, Georgia Institute of Technology, Atlanta, GA 30332, USA. [6] These authors contributed equally: Yufei Zhang, Zhihao Huo, Xiandi Wang. ✉email: wangxiandi@binn.cas.cn; cfpan@binn.cas.cn; zhong.wang@mse.gatech.edu

Wearable electronic devices that exploit wireless technologies offer simple, battery-free platforms for human-machine interaction, which plays an essential role in soft robotics, human healthcare monitoring, and implantable medical systems[1–7]. Existing commercial wireless wearable devices allow accurate monitoring of body movements, temperature, PH, blood pressure, oxygenation, and electrophysiology. Although these wireless devices have various capabilities, they are either rigid or flexible with specific structural designs that often fail to offer sufficient comfort, ease of use, and durability when attached to biological surfaces[8–13]. Hence, it is crucial to develop highly stretchable transparent materials to fabricate wireless wearable electronics with better performance. Recent advances in materials engineering[14–16], electronics[17–19], and technology[20–24], especially the development of self-powered systems based on the triboelectric and piezoelectricity[25–27], have led to the emergence of stretchable transparent devices that can conform to the complex, irregular surfaces of biological skin. However, high precision patterned stretchable transparent materials and efficient wireless power transfer under tensile state are the major limiting factors for the development of wireless electronics.

At present, several promising developments related to various wearable devices, such as smart lenses[28,29], sensors networks[30–33], and complex integrated circuits[34–38], combined with wireless technology have been shown to achieve real-time monitoring of human health. There are three main modes of wireless power transfer technology, namely magnetic induction, radiation, and resonant coupling. In particular, the magnetic resonant coupling mode takes precedence over the other modes in this study because it achieves a good trade-off between power transfer distance and efficiency, which has proven to be an important development since it was confirmed in 2007[39–42]. Nevertheless, few systematic studies have investigated wireless transmission systems under the tensile state or optimized device design to improve the power transfer efficiency. Moreover, stretchable transparent materials, which are the basis for the development of functional wireless devices, have been investigated intensively, including carbon nanomaterials[43–46], metal nanowires[47–50], conducting polymers, and stretchable metals with geometric designs[51–54]. Ionic conductors composed of hydrogels and conductive ions have attracted considerable attention owing to their ultra-stretchability and transparency[55,56]; however, their low conductivity and fluidity are major challenges for electrode patterning and device packaging. In particular, one-dimensional (1D) ultralong metal nanofibers have been prepared as percolation networks by electrospinning to fabricate stretchable transparent electrodes. Charge transport occurs along these 1D metal nanofiber networks, and their ultralong property facilitates reduction of the sheet resistance and enhancement of the stretchability by decreasing the number of junctions between the metal fibers. In our previous study[57–61], we have developed large-scale multifunction sensor networks based on triboelectric nanogenerator which is the application of the Maxwell's displacement current and is invented by Wang's group[62–64]. The devices present the wide ranges of superior resistance, high transparency and can detect random mechanical stress signals; hence, future works will focus on the design and preparation the various high precision patterned stretchable transparent electrodes to provide a powerful platform for wireless wearable electronics.

Here, we introduce a novel approach for fabricating transparent and stretchable wireless Ag nanofibers (Ag NFs) spiral coils combined with functional electronic devices for power transfer and information identification. Inspired by the two-dimensional ramified structure of leaves or silk, various high precision patterned Ag NFs electrodes can be fabricated using photolithography and wet etching, with the advantages of simple operation, low cost, and convenient large-scale preparation. Systematic research is conducted to improve the quality factor ($Q$) and radio frequency (RF) properties by optimizing the design scheme. Various cracks are generated in the Ag NFs electrodes under tensile strain, leading to changes in the $L$ and especially the $R_S$ of the coil. Basically, the $L$ of the coil is strongly affected by the dimension and layout owing to the coupling effect. Moreover, the working frequency of the device should be set below the self-resonance frequency $f_0$ or as the frequency with the maximum value of $Q$ for better performance. Based on the magnetic resonant coupling mode, we find that a five-turn stretchable spiral coil possesses 15% wireless power transfer efficiency with $\varepsilon = 100\%$. Furthermore, multifunction wireless electronics and signal detection systems are developed by integrating the Ag NFs spiral coils with other tiny electronic components. We realize short-distance content recognition by using NFC tags, as well as long-distance audio transmission by employing FM technology, thus providing a broad platform for smart wearable electronics.

## Results

**Fabrication and characterization of patterned Ag NFs coils.** Patterned Ag NFs electrodes were fabricated by photolithography and wet etching in an elastomeric polydimethylsiloxane (PDMS) matrix, as shown in Fig. 1a. Veins of leaves or silk structures in nature exhibit high transparency, excellent stretchability, and stability. Inspired by such structures, we developed Ag NFs nets with a core/shell structure similar to the two-dimensional ramified structure of leaves or silk. For device integration, it is necessary to overcome the problems of patterning of stretchable electrodes. The patterned Ag NFs electrodes in an elastomeric polydimethylsiloxane (PDMS) matrix can be designed via electrospinning, magnetron sputtering, photolithography, and wet etching (Supplementary Methods and Supplementary Fig. 1). For demonstration, a series of conductive designs were fabricated on a PDMS substrate (a peace dove, the letters "B", "I", "N", and "N", an office logo, and a pair of cartoon figures), as shown in Fig. 1b, c. Among them, the peace dove and the letters "B", "I", "N", and "N" were micro-patterned, where the clear edge of the Ag NFs could be observed and the line width was accurate up to several tens of micrometers. The office logo and cartoon figures were patterned with a large area and they exhibited high transparency. This method is suitable for preparing Ag NFs with various conductive patterns having promising applications in circuit systems.

The optimum structural design, including the coil structure, dimension, and conducting wire twining technique, significantly affects the quality factor ($Q$) of Ag NFs electrodes, defined as the ratio of the inductance to the AC resistance; which is an important parameter for wireless electronics. Actually, the skin effect, i.e., the current conduction along the outer surface of a conductor at high frequencies, will increase the effective resistance of the conductor. Nevertheless, for our as-synthesized patterned wireless coils, the resistance increase caused by the skin effect is negligible in this study owing to the core/shell structure of Ag NFs (Supplementary Notes 1, 2). The $Q$ value of straight Ag NFs electrodes was first investigated as the most basic structural unit, as shown in Fig. 1d, e. The inductance and resistance of straight samples was found to be positively correlated with their length and negatively correlated with their width, which is similar to the trend observed for copper wires. Notably, the Ag NFs electrodes have greater electrical resistance than commercial electrodes owing to their interconnected junctions. Then, the inductance of Ag NFs electrodes reduces from a straight shape to a shapely pattern due to the existence of mutual inductance among different line segments. Different shapes of the coils were

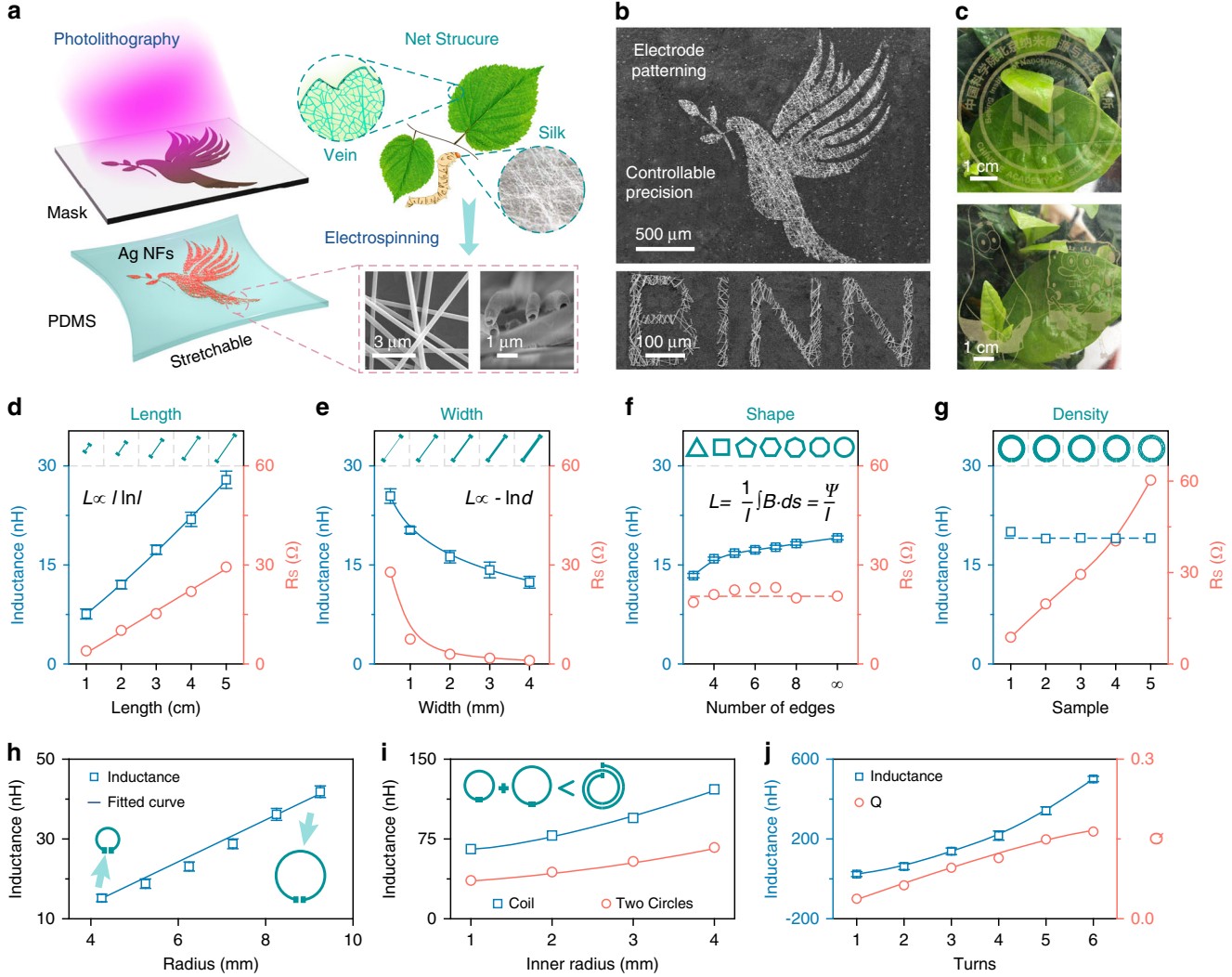

**Fig. 1 Fabrication and characterization of high precision patterned Ag nanofibers (Ag NFs) coil. a** Concept illustration the fabrication process of Ag NFs electrode with bioinspired net structure. **b** SEM images of high-precision patterned Ag NFs electrode. **c** Digital photos of large-area patterned electrode on PDMS substrate. Inductance and resistance of different electrodes with change in (**d**) length, (**e**) width, (**f**) shape, and (**g**) density. **h** Inductance variation of circular electrodes with radius. **i** Inductance comparison between two-turn spiral coil and two single-turn circular electrodes. **j** L and Q variation of electrodes with number of turns.

thus designed to further confirm the influence of geometry on inductance, which could also influence the magnetic field distribution near the Ag NFs electrodes. By maintaining the length of the coil, single-turn Ag NFs electrodes with different numbers of edges were designed to obtain different magnetic areas, as shown in Fig. 1f. Clearly, the circular electrode had the highest inductance, as its area was larger than that of the other polygonal electrodes, while all the coils showed similar resistance. Moreover, denser Ag NFs coils exhibit lower resistance but the same inductance (Fig. 1g). Hence, we inferred that the $R_S$ of the coil is strongly influenced by the dimension and material characteristics of the conducting wire, such as the length, width, and NFs density, whereas it is not affected significantly by the coil layout. Furthermore, for the single-turn Ag NFs electrode, the $L$ is mainly affected by the dimension and layout of the coil, especially the wire length (Fig. 1h, Supplementary Figs. 9–11). Finally, the coupling effect of multi-turn coils on the $Q$ value was explored, as shown in Fig. 1i, j. Two-turn spiral coils show higher inductance than two single-turn circular electrodes owing to the mutual inductance between them. As the number of turns of the spiral coil increases, the mutual inductance is enhanced correspondingly,

resulting in higher $L$ and $Q$ values (Fig. 1j). By analyzing the above-mentioned factors, including the straight and coiled shape, as well as the coupling effect, we can conclude that the $Q$ value strongly depends on the important physical parameters of the relevant structure. An optimized design scheme should be proposed to achieve a higher $Q$, i.e., the length and number of turns of the coil should be optimized for a stronger coupling effect.

**RF properties of Ag NFs coils under tensile strain.** The radio frequency (RF) properties of these Ag NFs electrodes were subsequently investigated when they were stretched; the $Q$ value was also influenced by the working frequency and strain variation. In this study, the scattering parameter $S_{11}$ of the device was obtained using the one-port scattering analysis method, as shown in Fig. 2a. An inductor lumped-parameter model consisting of inductance $L$, resistance $R$, and parasitic capacitance $C$ can be simplified into the frequency-dependent effective series resistance $R_S$ and reactance $X$ (Supplementary Fig. 12). Meanwhile, few fractures or cracks occur on the Ag NFs electrode owing to its high effective stiffness when the device is under strain, although the PDMS substrate with low effective stiffness can absorb most

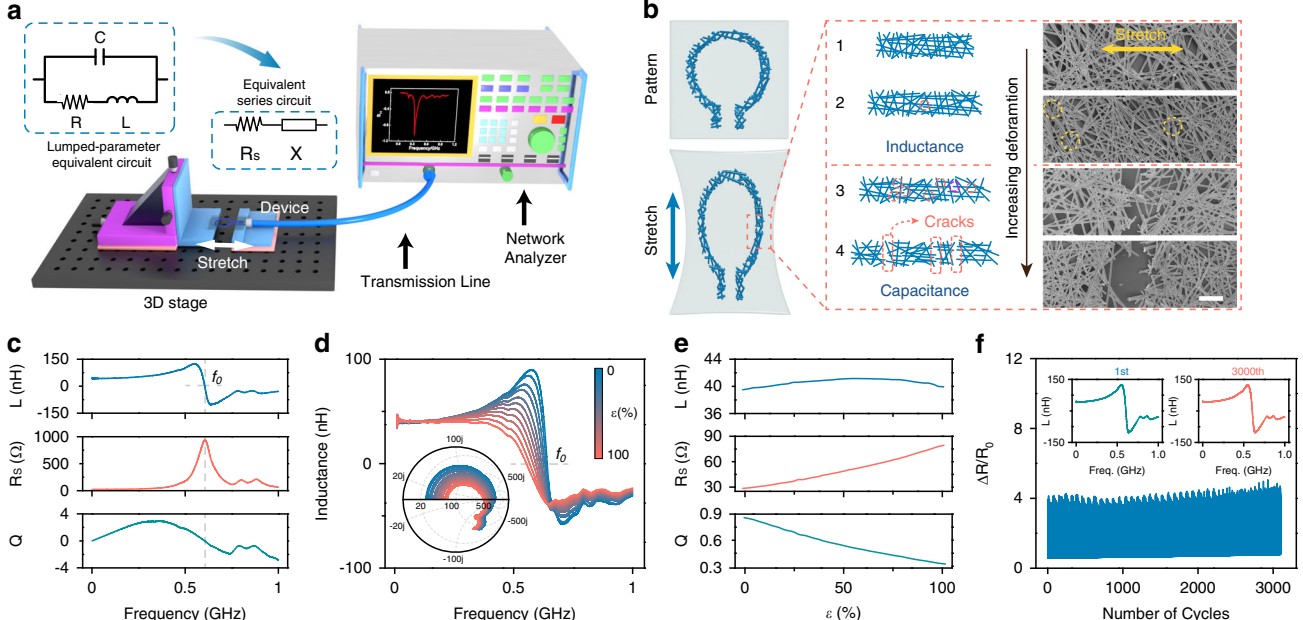

**Fig. 2 Radio frequency (RF) properties of the coil under tensile strain. a** Schematic of the experimental setup and equivalent inductance model of Ag NFs coil. **b** Schematic diagram and SEM images of the evolution of the Ag NFs under different tensile strains. **c** Inductance $L$, resistance $R_S$, and quality factor $Q$ versus frequency for single-turn coil shown in **b**. **d** Inductance variation versus tensile strain for the single-turn coil. Inset: Smith chart of single-turn coil. **e** Inductance $L$, resistance $R_S$, and quality factor $Q$ versus tensile strain at 10 MHz. **f** Stability measurement of single-turn coil. The cycle period is over 3000 cycles for $\varepsilon = 100\%$. Insets: inductance before and after cyclic endurance.

of the stresses via elastic deformation. Thus, the reactance characteristic of the coil changes, as shown in Fig. 2b. Within a small strain, the inductance is still dominant, while the parasitic capacitance increases with large fractures and cracks induced under high stress. In particular, the electrode loses its conductivity and is completely transformed into a capacitor when the cracks are sufficiently large. SEM images of the coils with different deformations are shown on the right of Fig. 2b (Supplementary Note 1), indicating that the cracks become more pronounced in the direction of the tensile strain.

The RF characteristic of the single-turn coil without tensile strain is shown in Fig. 2c. The $L$ of the coil first increased, then rapidly decreased, and finally became negative, which indicates a capacitor under high frequency. The self-resonance frequency $f_0$, defined as the point where the $L$ equals zero, was around 0.64 GHz, where the device also showed the maximum $R_S$ owing to the resonance oscillation. Moreover, the $Q$ reached its maximum value at around 0.36 GHz, which was the optimal frequency for this single-turn circular electrode. Subsequently, the change in the $Q$ of the circular electrode with the tensile strain was explored in the frequency range of 10 kHz to 1 GHz, as shown in Fig. 2d, e. It was found that the $f_0$ of the electrode shifted to lower frequencies (from 0.64 GHz to 0.55 GHz) as the tensile strain increased, whereas the $R_S$ increased (Smith chart, Fig. 2d (inset)). We inferred that the change in $f_0$ was mainly due to the variation of $R_S$ resulting from the external stress, which could be theoretically verified by the resonance frequency formula (Supplementary Note 3). For better understanding, the $L$ and $R_S$ as functions of the strain were measured at 10 MHz (Fig. 2e). Evidently, $L$ increased and then decreased, but it remained stable overall, while $R_S$ increased from 29 Ω to 77 Ω, resulting in a decrease in $Q$ from 0.85 to 0.32. According to the previous analysis of the reactance state of the coil under strain, it was speculated that the $L$ increased slightly with deformation owing to the change in the electrode length, whereas the coil would turn into a capacitor when the strain continued to increase owing to the enhancement

of the parasitic capacitance, especially for less dense devices (Supplementary Fig. 13). Hence, for this coil, the $R_S$ depending on the number of cracks was the key parameter to change the $Q$ and $f_0$ when the device was under strain. In addition, the coil showed excellent stability by repetitive tensile testing over more than 3000 cycles with $\varepsilon = 100\%$ (Fig. 2f). In other words, considering the effect of strain, it is better to select the working frequency below $f_0$ or at the frequency with the maximum value of $Q$, which is around 300 MHz for the single-turn coil. Both $R_S$ and $L$ of the Ag NFs coils will change under tensile strain owing to the generation of cracks. In particular, $R_S$ increases considerably with deformation, resulting in a decline in $Q$ to a certain extent.

The RF characteristic of the coil with different structural designs was investigated further to achieve an optimized coil structure that could work under tensile conditions, including the density of the Ag NFs, the shape of the electrode, and the coupling effect of multi-turn coils. First, we considered the material characteristic of the conducting wire, especially the density of the Ag NFs, which can also influence the coil transparency. By controlling the electrospinning duration, four coils with different Ag NFs densities were fabricated. The RF properties at 100 MHz under strain, as well as the transmittance spectra are shown in Fig. 3a. Apparently, the device with denser Ag NFs showed poorer transparency but smaller $R_S$ and more stable $L$, which means that the device exhibited higher conductivity and stretchability. We also inferred that appropriate enhancement of the NFs density is an effective way to prevent the device from turning into a capacitor (Supplementary Fig. 14). Second, the stress distributions of devices with various shapes are different when they are under stress. We found that the orientation of the NFs will not affect $L$ by measuring the change in $L$ of straight electrodes with single-orientation NFs as a function of the stretching angle with $\varepsilon = 100\%$, as shown in the inset of Fig. 3b. Nevertheless, large inductance differences existed for devices with square or circular coils under the same stress, and the $L$ of the circular coil was more stable under different strains.

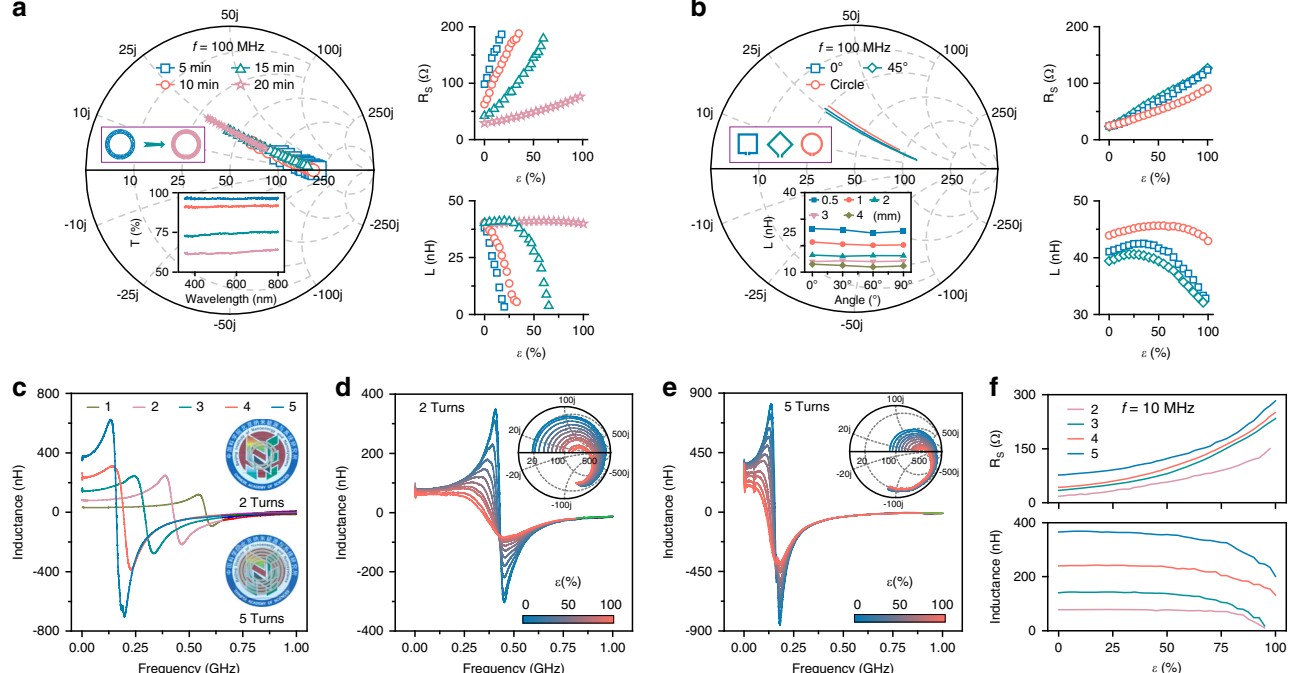

**Fig. 3 Optimization of coil structure under tensile strain. a** Smith chart of tensile properties for single-turn coil with four electrospinning durations. The inductance $L$ and resistance $R_S$ are shown on the right. Insets: UV-visible spectra of Ag NFs with four electrospinning durations. **b** Smith chart of tensile properties of coils with different shapes. The inductance $L$ and resistance $R_S$ are shown on the right. Insets: inductance $L$ of unidirectional Ag NFs with different orientations. **c** Inductance variation versus frequency of multi-turn spiral coils. Insets: photographs of the two-turn and five-turn spiral coils. Inductance variation versus tensile strain for (**d**) two-turn and (**e**) five-turn spiral coils. Insets: Smith chart of two-turn and five-turn spiral coils. **f** Inductance $L$ and resistance $R_S$ of multi-turn spiral coils versus tensile strain at 10 MHz.

This is because more cracks would be generated at the corners, where there is greater stress (Supplementary Fig. 15). Hence, the circular coil is more appropriate for stretchable wireless electronics. Finally, the RF properties of multi-turn coils with or without tensile strain were obtained, as shown in Fig. 3c–f. With no stress, $f_0$ shifted rapidly to lower frequencies as the number of turns increased owing to the considerable enhancement of $L$ (Fig. 3c). Meanwhile, $f_0$ moved slightly toward lower frequencies for each multi-turn coil under strain owing to the significant increase in $R_S$ (Fig. 3d–f, Supplementary Fig. 16). It was assumed that the $R_S$ of each multi-turn coil was mainly influenced by the value of the stress, while the $L$ of the coil was dominantly affected by the number of turns owing to its stronger coupling effect. Therefore, we need to consider the effect of strain on $R_S$ and improve the material properties to reduce it, while $L$ should be designed according to the requirements of practical applications.

To further understand the variation of Ag NFs network under tensile strain, Ag NFs with different orientations (unidirectional, bidirectional, random) were simulated via finite element analysis method, as shown in Fig. 4a. The orientation of Ag NFs has a major impact on the stretchable property of the electrode. For the unidirectional Ag NFs network, the nanofibers have a significant stress concentration when the orientation of Ag NFs is in the similar direction of the tensile strain, especially at the interconnected junction. Similarly, the nanofibers along the tensile direction are still in a high-stress state for the bidirectional Ag NFs, while it is relatively lower in another direction. It is worth mentioning that the Ag NFs network with random structure has the lowest stress concentration, which means it is more likely to withstand higher strain. The corresponding SEM images of these Ag NFs networks under strain are shown in the Fig. 4b (more analysis could be seen Supplementary Fig. 17 and Supplementary Note 5), indicating that the multi-orientation nanofibers possess various conductive paths even under high strain. In addition, Fig. 4c

illustrates the stress distribution of different patterned Ag NFs electrodes (square, hexagonal and circular) under deformation. It is found that the electrode edges along the stretching direction have greater stress. And it is easier to generate the stress concentration at the corners of the patterned electrode where the large cracks are also more likely to develop with the increasing strain, as shown in Fig. 4d (detailed information can be seen in Supplementary Figs. 18–21). Therefore, circular electrodes based on the random orientation Ag NFs will perform the better tensile property and it's an ideal structural design for stretchable electronics.

**Wireless power transfer efficiency.** To satisfy the requirements of practical applications, it is essential to explore the wireless transmission capability of the stretchable spiral coils coupled with external coils. As a new concept of energy transmission, wireless power transfer technology consists of three main modes: magnetic induction, radiation, and resonant coupling. Typically, the magnetic resonant coupling mode takes precedence over the other modes in this study because it achieves a good trade-off between power transfer distance and efficiency. Figure 5a shows the magnetic field distribution simulated by the finite element analysis method (COMSOL) under high-frequency excitation, as well as the T-type equivalent circuit based on the two-port network theory. The wireless power transfer efficiency $\eta$ can be deduced as follows:

$$\eta = \frac{|S_{21}|^2/Z_0}{\Re\left[|S_{11}+1|^2/Z_{\text{in}}\right]} \qquad (1)$$

where $S_{21}$ is the forward transmission coefficient, $Z_0 = 50\,\Omega$ is the characteristic impedance of the measurement system, and $Z_{\text{in}}$ is the input impedance of the device under testing (Supplementary Fig. 22, Supplementary Note 6). Notably, the mutual inductance $M$ is the key parameter in this wireless power transfer (WPT)

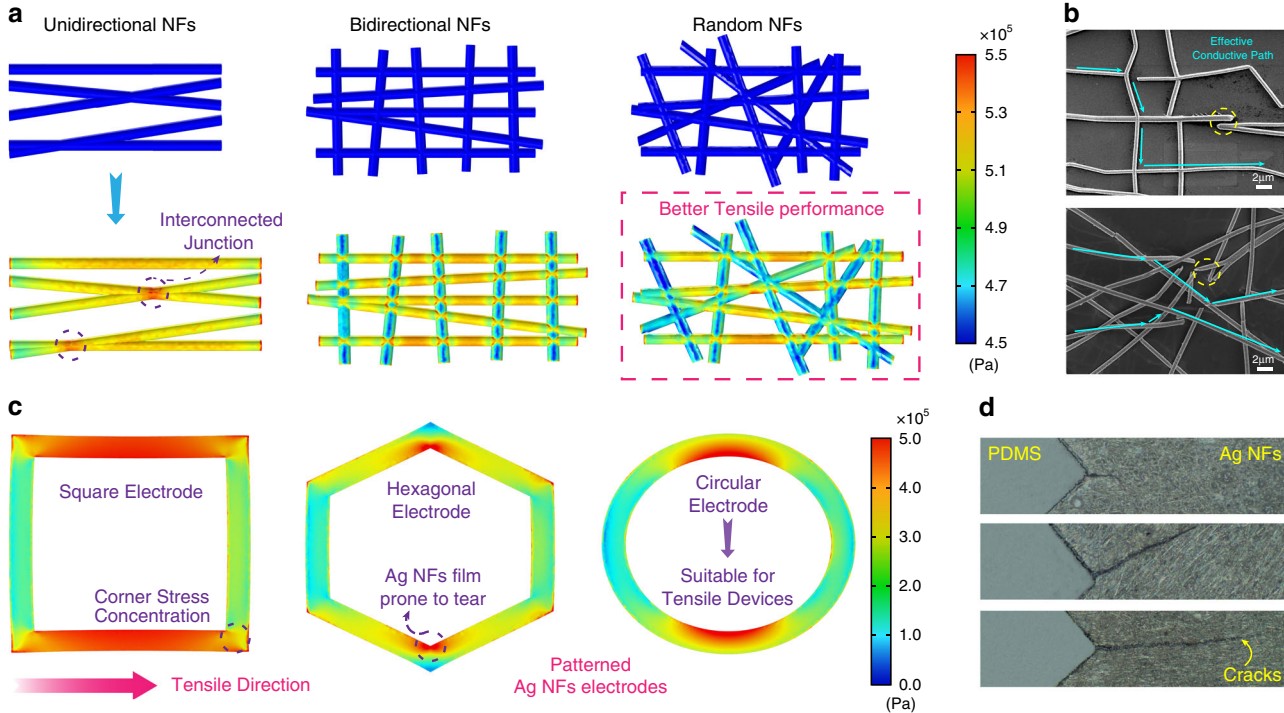

**Fig. 4 Simulations of the mechanical properties of various Ag NFs networks. a** Stress distribution of Ag NFs networks under tensile strain with different orientations (unidirectional, bidirectional, random). **b** SEM images of the Ag NFs networks under tensile strain (Top: bidirectional NFs, bottom: random NFs). The NFs break under stress, but there are still conductive paths in the electrodes. **c** Stress distribution of different patterned Ag NFs electrodes (square, hexagonal and circular) under strain. Stress concentration tends to occur in the corners of the electrodes. **d** Optical images of large cracks at the corners of square electrodes with the increasing strain (Top to bottom: 15%, 45% and 60%).

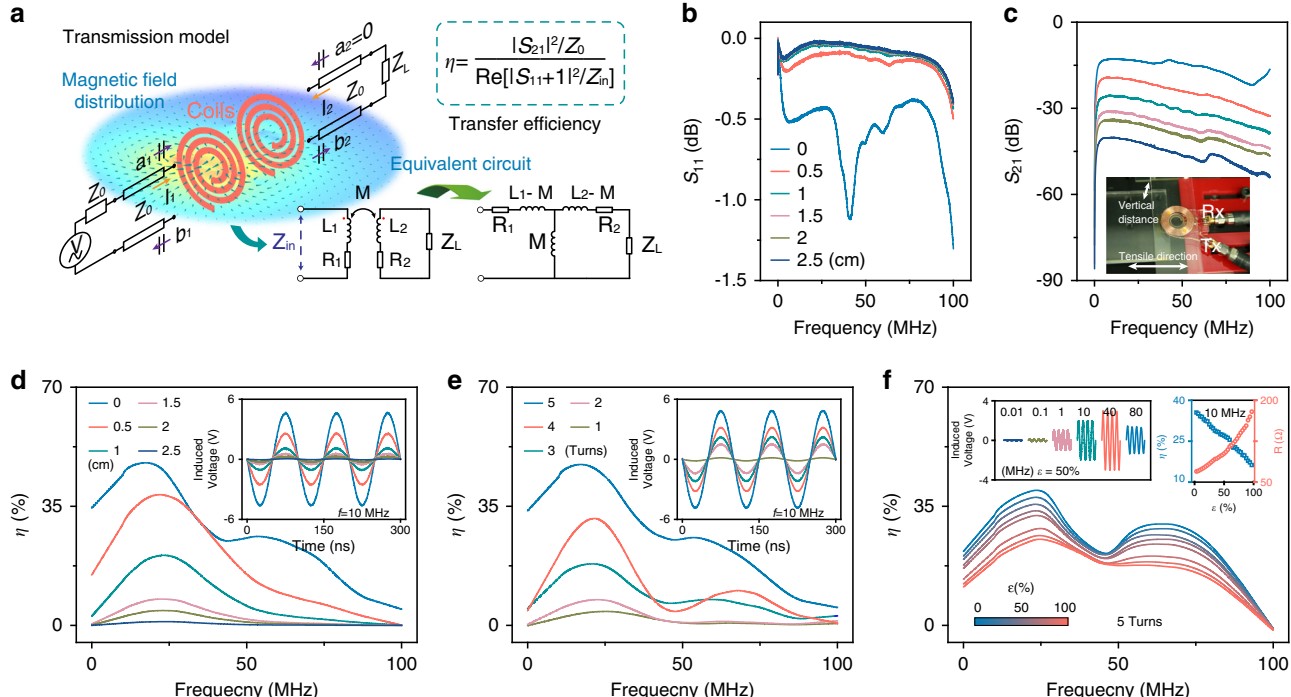

**Fig. 5 Wireless power transfer efficiency of stretchable spiral coils. a** Two-port network model analysis based on magnetic resonant coupling mode. Scattering parameters $S_{11}$ and $S_{21}$ and efficiency $\eta$ (**b**–**d**) between primary coil and stretchable coil at different vertical distances. Insets: photographs of coupling of two coils under testing (**c**); induced voltage of two coils with different vertical distances at 10 MHz (**d**). **e** Efficiency $\eta$ between primary coil and stretchable coils with different numbers of turns. Insets: induced voltage shown in **e** at 10 MHz. **f** Efficiency $\eta$ between primary coil and stretchable coils with tensile strain. Insets: induced voltage measured from stretchable coils with a tensile strain of 50% at different frequencies (left); efficiency $\eta$ and resistance $R_S$ versus tensile strain at 10 MHz (right).

system, which can change the electrical parameters of the equivalent circuit and thus affect its $\eta$. Therefore, we concentrate on the influence of various factors on $M$ when the device is under strain, including the relative geometric position of two coils and the spiral coil parameters. Basically, $M$ decreases as the spacing between two stretchable spiral coils increases. The scattering parameters as functions of the vertical distance in the frequency range of 10 kHz to 100 MHz are shown in Fig. 5b, c. As the vertical distance decreased, the $S_{11}$ parameter decreased whereas the $S_{21}$ parameter increased, indicating that more energy in the primary coil was transferred to the secondary coil. Further, $\eta$ under different frequencies was calculated using the above-mentioned formula, as shown in Fig. 5d. As expected, the transfer efficiency increased significantly as the distance decreased. It should be noted that the highest power transfer frequency is approximately 20 MHz, which depends on the inductance and parasitic capacitance of the primary coil (Supplementary Figs. 24, 25). In particular, we obtained the induced voltage between two coils with different perpendicular distances at 10 MHz, as illustrated in the inset of Fig. 5d. Meanwhile, the number of turns of the stretchable spiral coils also influences $M$ (Fig. 5e). The greater the number of turns of the coil, the higher is the efficiency and induced voltage. In addition, the effects of different lateral distances and rotation angles on $M$ were also investigated (Supplementary Fig. 26a, b). Finally, $M$ slightly changed under tensile strain owing to the change in the $L$ and $R_S$ of the coil, as depicted in Fig. 5f. Clearly, the $R_S$ of the coil increased under tensile strain, as discussed earlier, resulting in a significant drop in transfer efficiency. Specifically, the efficiency dropped from 35% to 15% at 10 MHz when the strain increased to 100% (five-turn stretchable spiral coil, Supplementary Fig. 26c). At $\varepsilon = 50\%$, the induced voltage at different frequencies was measured. We found that the voltage increased with the frequency, but it dropped at 80 MHz owing to the self-resonance frequency of the primary coil. Accordingly, we believe that the stretchable transparent spiral coil exhibits excellent wireless transmission capability even in the tensile state, and the wireless power transfer efficiency can be improved by setting the appropriate distance, operating frequency, and electrical parameters of the coil.

**Wireless power transfer and signal detection system.** The stretchable transparent Ag NFs spiral coil can be integrated with other tiny electronic components (1.6 mm × 1.2 mm × 0.45 mm, Supplementary Fig. 27) and assembled into more complex functional wireless electronics for power transfer and data communication. Figure 6a shows a typical device structure that is easy to operate. PDMS was employed as the substrate, the insulator layer, and the encapsulation layer, which were transparent and stretchable to absorb the strain energy. Tiny electronic components with high effective stiffness adhered to the stretchable substrate, which could withstand less stress when the device was under strain on the basis of the island-bridge design mode[65]. A wireless light-emitting diode (LED) module was fabricated using lithography and three-dimensional printing technology to demonstrate the wireless power transmission capability, as shown on the right of Fig. 6a. The device could also work in water or a high-moisture environment (details can be found in Supplementary Movie 1). We believe that this technology is promising to develop stretchable electronics for adverse conditions.

To achieve data communication, we designed two devices using near-field communication (NFC) and frequency modulation (FM) technology. First, a thin NFC chip (NTAG 213) was employed on the basis of the ISO/IEC 14443 Type A standard instead of the LED module described above. Figure 6b shows the NFC tag block diagram of the functional components. The Ag NFs spiral coil tags

can operate in a battery-free mode via an external reader (i.e., any NFC-enabled smartphone, tablet, or watch), with data and power transmission by magnetic inductive coupling based on NFC protocols. In this study, the coil was redesigned with 2.7 μH and 232 Ω at 13.56 MHz (Supplementary Fig. 28, Supplementary Note 7). Images of a representative device and its application (identification of tortoise A) are shown in Fig. 6c, d.

Frequency modulation (FM) refers to the encoding of information in a carrier wave by varying the instantaneous frequency of the wave, which is suitable for long-distance communication. As shown in Fig. 6e, f, the audio signal (music or sound) is modulated onto the carrier wave via an FM generator, which is then received by a stretchable spiral coil and fed into a spectrum analyzer for auditory and visual monitoring. Here, we mainly studied the effects of the tensile state and transmission distance on information resolution. A simple piano music audio input was provided to the above-mentioned system, and the acoustic waveforms and spectrograms were analyzed by real-time fast Fourier transform, as shown in Fig. 6g (Supplementary Movie 2). Under different tensile strains, the harmonic frequency of each note could be obtained correctly. More complicated data, such as human voice, was tested, where different speakers repeated four simple letters ("B", "I", "N", "N") in front of a standing microphone. Figure 6h shows the typical male and female auditory spectrum. Evidently, different letters had different frequency distributions and the female spectrum was slightly higher than the male spectrum (Supplementary Fig. 29). Another test was conducted to confirm that the stretchable spiral coil could successfully pick up audio signals at different distances. As shown in Fig. 6i, the signal at low frequency could still be distinguished as the distance increased, while noise gradually appeared in the high-frequency region.

## Discussion

The high-precision epidermal radio frequency antenna via Ag NFs network is suitable for wireless stretchable multifunction electronics, which is promising to achieve a wide range of application in human-machine interfacing, health monitoring and intelligent electronic skin. Nowadays, various wireless wearable devices allow accurate monitoring of body movements, temperature, PH, blood pressure, oxygenation, and electrophysiology, just like smart bracelet, smart phone and wearable electronics. Here, high-precision epidermal electronics provide multifunctional services for human and offer sufficient comfort, ease of use, and durability when attached to biological surfaces. However, the Ag NFs spiral coils possess the large resistance and it is difficult to combine with the impedance matching circuit. Hence, reducing electrode resistance and designing better impedance matching networks can be considered to improve the performance of the device in the future work.

In summary, we presented a highly stretchable transparent spiral coil antenna based on Ag NFs for wireless power transfer and information identification. Inspired by natural systems such as veins of leaves or silk, various high-precision patterned Ag NFs electrodes with a net structure were fabricated using photolithography and wet etching, with the advantages of simple operation, low cost, and convenient large-scale preparation. An optimized design scheme was shown to be the key to obtaining higher $Q$ and RF properties of the coil. When the device was under strain, various cracks were generated in the Ag NFs, resulting in changes in the $L$ and especially the $R_S$ of the coil. By using the coupling effect, the $L$ can be significantly enhanced to meet the requirements of practical applications. Considering the effect of strain, the working frequency of the device should be set below the self-resonance frequency $f_0$. For instance, a value of

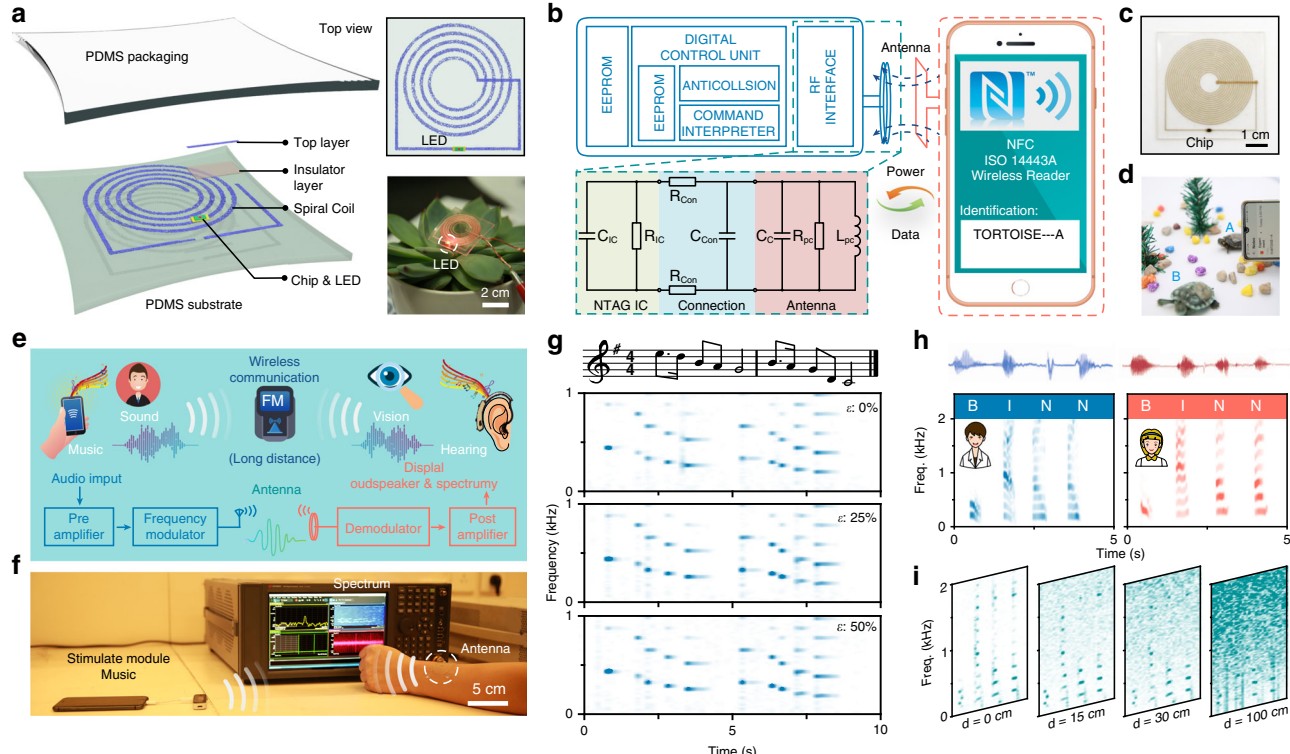

**Fig. 6 Functional wireless electronics for power transfer and data communication. a** Schematic illustrating the exploded view of the stretchable transparent devices. The top view and wireless powered devices are shown on the right. **b** Functional block diagram of near-field communication (NFC) tags with a smartphone that includes an NFC reader. **c** Digital photo of the NFC tags. **d** NFC tags attached to tortoises for identification. **e** Concept illustration of wireless signal detection system. **f** Photograph of measurement systems. **g** Detection of piano music audio signal with the coil under different tensile strains. **h** Detection of male and female human voice spectrum with simple letters ("B", "I", "N", and "N"). **i** Spectrum variation with measurement distance.

around 300 MHz is suitable for a single-turn coil. On the basis of the magnetic resonant coupling mode, we found that the coil showed outstanding wireless transmission capability even in the tensile state, which was approximately 15% at 10 MHz for a five-turn stretchable spiral coil with $\varepsilon = 100\%$. Finally, various types of wireless power transfer and signal detection systems were developed by integrating the Ag NFs coil with other tiny electronic components. We realized short-distance content recognition by using an NFC tag and long-distance audio transmission by employing FM technology. In conclusion, the proposed device has considerable potential for application in information identification systems, soft robotics, and wearable electronics.

## Methods

**Preparation of Ag NFs with core/shell structure**. PVA aqueous solution with a concentration of 10 wt% was prepared by mixing PVA powder (molecular weight, 80000) with deionized water. The mixture was stirred at 500 r/min at a temperature of 30 °C for 24 h using a magnetic stirrer to obtain a homogeneous solution. Subsequently, PVA NFs with random orientations could be fabricated using electrostatic spinning (SHENZHEN TONG LI TECH Co. Limited). Then, the PVA aqueous solution was loaded into a 10 ml commercial syringe with a polished G7 needle. The flow rate of the solution in the syringe pump was controlled at 0.4 ml h$^{-1}$, and a constant potential of 13 kV (negative voltage of 4 kV and positive voltage of 9 kV) was employed between the needle and the grounded substrates (acrylic frame covered with aluminum foil) at a distance of 10 cm. The air humidity was maintained within 25%. PVA NFs with an average diameter of 450 nm could be obtained, and a layer of silver (≈200 nm) was plated onto the surface of the PVA NFs via magnetron sputtering (PVD75 Kurt J. Lesker, Ar, 4 mTorr, 100 W, 15 min) to form NFs with a uniform dense core/shell structure.

**Fabrication of high precision patterned Ag NFs network**. The PDMS (SYL-GARD184 Dow Corning) main agents and curing agents (10:1 by weight) were mixed thoroughly and then degassed for 5 min to remove air bubbles before spinning and drying on the glass substrate. The PVA/Ag NFs were transferred to the surface of deionized water and the prepared PDMS substrate was used to pick

up the Ag NFs. Subsequently, a heat blower was used to dry the Ag NFs in a certain direction and ensure strong adhesion to the substrate. A negative photoresist (SUN-9i, 50 cp, SUNTIFIC) was spin-cast onto the surface of the Ag NFs at 2000 rpm for 60 s, and various mask layers could be obtained through a series of UV lithography steps (MA6 SUSS). Different masks were fabricated by laser direct systems (Heidelberg Instruments DWL 66+). Subsequently, Ag NFs without photoresist protection were etched with HNO$_3$ (5 M), during which the solution was stirred to eliminate the generated bubbles, and the remaining photoresist was cleaned with acetone. Finally, the patterned Ag NFs were packaged by another PDMS layer to prevent damage.

**RF properties characterization and measurements**. The optical transparency of the Ag NF electrodes was measured by a UV-vis-NIR spectrophotometer (UV-3600 SHIMADZU). The morphologies of the as-synthesized Ag NFs were characterized via a field-emission scanning electron microscope (SU8020 Hitachi) and a precision impedance analyzer (WK6500B) for measuring the inductance and resistance of the electrode. A stepping motor (LinMot E1100) was used to measure the mechanical stability of the sample. As for the RF properties, the Ag NFs coils were tested via microwave scattering analysis using a vector network analyzer (R&S ZNC3). The Ag NFs coils were characterized by one-port scattering analysis. The scattering parameter $S_{11}$ was obtained to be equivalent to the reflection coefficient of the device under testing, and the input impedance $Z_{in}$ was evaluated according to the following formula:

$$Z_{in} = Z_0 \frac{1 + S_{11}}{1 - S_{11}} \qquad (2)$$

where $Z_0 = 50\ \Omega$ is the characteristic impedance of the vector network analyzer and transmission cable, and $Z_{in}$ is a function of the frequency that represents the radio frequency characteristics of the Ag NFs coils. Further, the $L$ and $Q$ values could be extracted from $Z_{in}$ by the following relations:

$$L = \frac{\Im[Z_{in}]}{\omega} \qquad (3)$$

$$Q = \frac{\Im[Z_{in}]}{\Re[Z_{in}]} \qquad (4)$$

where ω is the angular frequency. Two-port microwave scattering analysis was used to represent the wireless power transfer efficiency, where η could be deduced using

the two-port scattering matrix (including $S_{11}$, $S_{12}$, $S_{21}$, and $S_{22}$). The frequency modulation module RC-127 was used for modulating the audio signals, and a spectrum analyzer (N9030B Keysight) was used to detect the signals wirelessly.

**Reporting summary**. Further information on research design is available in the Nature Research Reporting Summary linked to this article.

## Data availability

All the data supporting the findings of this study are available within the main text and the Supplementary Information. Data are also available from the corresponding authors upon request. Source data are provided with this paper.

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

## Acknowledgements

The authors express thanks for the support of National Natural Science Foundation of China (Nos. 61804011, 51622205, 61675027, 51432005, 61505010, 51502018), national key R & D project from Minister of Science and Technology, China (2016YFA0202703), Beijing City Committee of Science And Technology (Z171100002017019 and Z181100004418004), Natural Science Foundation of Beijing Municipality (2184131, 4181004, 4182080, 4184110, and Z180011), and the University of Chinese Academy of Sciences.

## Author contributions

Z.L.W., C.F.P., X.D.W., Y.F.Z., and H.Z.H. conceived the idea. C.F.P., X.D.W., Y.F.Z., and H.Z.H. designed the experiments. Y.F.Z., H.Z.H., X.D.W., X.H., W.Q.W., B.S.W., and J.T. performed the experiments and analyzed the data. Y.F.Z., H.Z.H., X.D.W., H.W., J.Y.Z., C.F.P., and Z.L.W. wrote the paper. All authors discussed the results and commented on the manuscript. Y.F.Z., H.Z.H., X.D.W., contributed equally to this work.

## Competing interests

The authors declare no competing interests.
