## [Peer Review File · Nature Communications]

Reviewers' comments:

Reviewer #1 (Remarks to the Author):

Review

High-precision epidermal radio frequency antenna based on nanofiber network for wireless stretchable multifunctional electronics

Zhang and co-authors address the fabrication method and characterization of transparent and stretchable wireless Ag nanofiber (NF) coils, which are inspired by silk structures in nature. Characterization of patterned, stretchable coil is a significant step in certain applications, especially epidermal wireless health monitoring devices. The contributions in terms of RF properties of different types of designs are important, and the sections providing details on the results and discussion of those aspects are strong. There are several sections of the paper that are lacking in terms of application of the proposed device. Below are some comments and suggestions for improvement of those sections.

Comments

- i. The analysis in Fig.1 concerning the relationship between electrical and geometrical parameters is very interesting as well as the fatigue test in Fig. 2. However, it would be more powerful to define a close-form relationship (it could be derived by interpolating the data in the shown charts) between the geometrical and electric (resistance and inductance) parameters in order to have better control of the structural characterization.
- ii. In Fig.2, the SEM images appear to have been taken in different spots, so it is difficult to compare cracks caused by deformation and the intrinsic structural gap after fabrication.
- iii. All deformation is limited under $\epsilon = 100\%$. More details should be included about the yield point to indicate that the $\epsilon = 100\%$ is an elastic region.
- iv. From an electromagnetism viewpoint, the geometrical characteristics of circular coil's self-inductance is already very well known. So, Fig.3, 4, and 5 appear to be a validation of coil design rather than giving a new finding.
- v. Other important parameters for AC resistance of conductors including permeability and conductivity of the medium is never mentioned. Please specify permeability and conductivity of the medium.
- vi. Authors discuss the stretching capabilities of the presented Ag NF coil, but a mechanical analysis has not been shown. It would be useful to model the coil to validate how the mechanical stress impacts its structure and if there are some critical/weak points. Moreover, for a wearable device, the mechanical stress is related to the body part where the device is applied itself, hence a crucial role is played by the design of the coil. In order to keep the antenna parameters as robust as possible while undergoing deformations (Fig. 3 shows how the deformation affects the inductance and the resistance) an example of ad-hoc design for a specific human body part application should be shown (e.g. the tensile stretch caused by the neck movement will be different from the one coming from an arm, as shown by C. Miozzi et al in "Radio-mechanical Characterization of Epidermal Antennas during Human Gestures").
- vii. In the last section, authors show a possible application of the antenna involving Frequency Modulated (FM) communication. In the mentioned test, there is lack of information about the carrier frequency. In this scenario, it would be interesting to provide a measurement of the IEEE Gain of the antenna, with an estimation of the maximum reading distance with a 4 W EIRP condition.
- viii. Give specific descriptions instead of using vague wording - line 104, "excellent wireless power transfer efficiency": it should be careful to say that 15% power transfer efficiency (PTE) is excellent without reference. Please add appropriate reference to clarify this sentence.
Line 336, "integrating the Ag NF coil with tiny electronic components": please specify the size of components for clarity.

These aspects of the paper can be improved greatly and this would lead to a much stronger publication.

Reviewer #2 (Remarks to the Author):

Zhang and coauthors demonstrated to fabricate High-precision epidermal radio frequency antenna based on nanofiber network for wireless stretchable multifunction electronics. Ag nanofiber (NF) is attracting much attention because it has very high conductance and high transparency, which cannot be achieved with conventional metals. I understand the importance of the Ag NF and it will open the new era of flexible and stretchable electronics.

However, I could not understand the novelty of this work. For example, authors described "Notably, the Ag NF electrodes have greater electrical resistance than commercial electrodes owing to their interconnected junctions." But I could not understand why the Ag NF based electrodes has the electrical characteristics greater than commercial electrodes. Authors should carefully explain the mechanism of the interconnected junctions. This manuscript does not describe the microstructure and electrical characteristics with stretching and bending. At least, authors should explain the following questions and contain the experimental data for showing the novelty of the work and for better understanding of the proposed materials and the antenna.

1. The micrographs of the antennas with stretching and/or bending should be contained for showing the changes in the nanofiber networks.
2. Describe what is the new compared with conventional Ag nanowire-based electrodes. If interconnected junctions is superior to the conventional nanowires, please explain the reasons. For example, Supplementary Figure 5 shows the SEM images of Ag NFs under tensile strain, but does not contain the changes in the situation of junctions. For better understanding of the novelty, data of the junctions is indispensable.
3. The demonstration movies are not new and does not contain the scientific novelty of the work. No stretching and bending in the movies.

In summary, although Ag NF is one of the most interesting materials toward new flexible and stretchable electronics, this paper only shows the characteristics of an antenna made of nanowires, but does not give any information on the nanostructure behind it or any structural characteristics when the coil is deformed. Authors should carefully compare their work with pioneering works related to Ag nanowires and then describe the novelty of the Ag NF based antennas. Comparison table related to nanofiber-based antenna may be effective.

Reviewer #3 (Remarks to the Author):

This manuscript entitled 'High-precision epidermal radio frequency antenna based on nanofiber network for wireless stretchable multifunction electronics' reports fabrication of flexible, wireless devices using Ag nanofiber spiral coils. The nanofibers are processed through lithography and wet etching to create several patterns. The authors have carried out significant investigation on the properties of the coil, including Quality factor and Radio frequency by considering the effects of strain on resistance and inductance of the coils. The novel synthesis technique combining electrospinning, lithography and wet-etching empowers the authors to synthesize precise electrodes which can be investigated for crack induction and its consequent effects on transmission capabilities. Overall, this manuscript is well written and experiment process is neatly organized. Especially, by analyzing the different shapes, turns and length of the coils which are important factor for influencing the magnetic field, and consequently the wireless transmission capability via the magnetic resonant coupling mode, the authors present a holistic evaluation of the coils and the synthesized devices. The reviewer thinks that this manuscript could give insights for the future of precision synthesis of conductive electrodes, and inspire efficient combinations of techniques to advance flexible electronics. Therefore, as a reviewer, I recommend that this manuscript can be published in Nature Communications after Minor revision.

1. In the Introduction part, additional references related to the magnetic resonance coupling transmission mode should be added to give more background to the reader.
2. Since the durability of these flexible devices is an important aspect of the whole synthesis process, a measure for the adhesion of the nanofibers on the substrate would be a valuable addition to the manuscript.
3. Several references can be added to help the readers to understand previous works related to this manuscript:
 - A. Advanced Science 5 (11), 1801146.
 - B. Advanced Functional Materials 27 (29), 1701138.

Point to Point Response to the referees' reports
(comments in black, responses in blue, changes highlighted in yellow):

Reviewer #1:

Zhang and co-authors address the fabrication method and characterization of transparent and stretchable wireless Ag nanofiber (NF) coils, which are inspired by silk structures in nature. Characterization of patterned, stretchable coil is a significant step in certain applications, especially epidermal wireless health monitoring devices. The contributions in terms of RF properties of different types of designs are important, and the sections providing details on the results and discussion of those aspects are strong. There are several sections of the paper that are lacking in terms of application of the proposed device. Below are some comments and suggestions for improvement of those sections.

Answers:

We like to express our sincere thanks to the referee for her/his great effort to review the manuscript and positive evaluation on our work.

1. The analysis in Fig.1 concerning the relationship between electrical and geometrical parameters is very interesting as well as the fatigue test in Fig. 2. However, it would be more powerful to define a close-form relationship (it could be derived by interpolating the data in the shown charts) between the geometrical and electric (resistance and inductance) parameters in order to have better control of the structural characterization.

Response:

Thanks the reviewer for the suggestion. It is essential to figure out the relationship between the geometrical and electrical parameters to satisfy the practical applications, as the reviewer mentioned. The as-synthesized Ag NFs coils are different from the conventional thin film electrodes in morphology and structure, with the properties of network and core-shell structure. So, it is the focus of this work to find out how the optimum structural design significantly affects the quality factor of Ag NFs electrodes.

In this work, the **straight** Ag NFs coil was first investigated as the most basic structural unit. Then, the inductance of the coil reduces from a straight shape to a shapely pattern due to the existence of mutual inductance between different line segments. These relevant detailed qualitative description and experimental data are described in the revised MS (**Supplementary Note 3**). On the other hand, for the stretchable electrodes, the material properties (including NFs density and diameter, etc.) may also have a certain impact on the electrical parameters. So, it's difficult to give a common formula suitable for all electrodes.

Here, we illustrated the rules of corresponding geometrical factors on the inductance in the MS and the research approach was shown in **Supplementary Figure 11**. Additionally, as the reviewer suggested, for better understanding, **we further quantitatively analyzed the relationship between the geometrical and electric parameters** with the Ag NFs coils frequently used in this work (sheet resistance is 11.1~1.68 Ω/\square at >70% transmittance). *An estimation formula was summarized as following:*

$$L \approx L_{straight} - 1.4998 \times 10^{-4} \times \Delta S - 5.203 \times 10^{-10}$$

where $L_{straight} = 2 \times 10^{-7} \times l \times [\ln(l/d) - 0.912]$.

The main text and the Supplementary Note are revised according to your comment, which is listed as following for your convenience (all changes made in the revised MS is highlighted in yellow):

Main text, Page 7

Notably, the Ag NFs electrodes have greater electrical resistance than commercial electrodes owing to their interconnected junctions. Then, the inductance of Ag NFs electrodes reduces from a straight shape to a shapely pattern due to the existence of mutual inductance among different line segments. Different shapes of the coils were thus designed to further confirm the influence of geometry on inductance, which could also influence the magnetic field distribution near the Ag NFs electrodes. By maintaining the length of the coil, single-turn Ag NFs electrodes with different numbers of edges were designed to obtain different magnetic areas, as shown in Fig. 1f.

Supplementary Figures, Supplementary Note 3, Page S12-S16

Supplementary Note 3: Theoretical calculation of Ag NFs electrodes inductance

(a) Theoretical analysis

The electrical performance of Ag NFs electrodes is mainly affected by the inductance and resistance. Resistance is an important parameter for electrode quality factor. From the perspective of electrode dimension, the electrode length or width will affect the resistance of electrode. On the other hand, NFs density or the sputtering duration also affect the resistance of electrode in terms of the material characteristic. Hence, it is essential to reduce the Ag NFs electrode resistance for the higher quality factor by improving the preparation process.

Inductance is another important parameter for the radio frequency characteristics of the Ag NFs coils, which is mainly affected by three aspects (length and width, shapes and coupling effect).

i) Straight electrodes with various length and width

According to Supplementary Figure 9a-b, the relationship of the straight electrode between the length or width and the inductance can be expressed:

$$L = \frac{\mu_0 l}{2\pi} [\ln(l/d) - 0.75] \quad (3-1)$$

μ_0 : permeability of free space, $4\pi \times 10^{-7}$ H/m

l : length of straight electrode

d : width of straight electrode

Supplementary Figure 9 | Fitting of the straight electrode with inductance variation versus the increment in length (a) and width (b).

ii) Single-turn electrodes with various shapes

For single-turn electrodes with different shapes, the theoretical calculation of inductance is shown below:

$$L = N - G + A - Q \quad (3-2)$$

N : parameter determined only by the shape and size of the wire axis;

G, A, Q : parameters determined by the shape and size of the wire cross section and the current distribution in the wire cross section. The A and Q could be ignored in this article due to the small amounts.

$$L = N - G \quad (3-3)$$

Regular polygon electrode inductance (N):

$$N = \frac{\mu_0 a n}{2\pi} [\ln a + f(n)] \quad (3-4)$$

n : Number of sides of the polygon

a : Length of one side

$f(n)$: Parameter related to the number of sides of the polygon shown in Table 4

Circular electrode inductance (N):

$$N = \mu_0 R [\ln 8R - 2] \quad (3-5)$$

R : Radius of the circular electrode

$$G = \frac{\mu_0 l}{2\pi} [\ln(d/2) - 0.0584] \quad (3-6)$$

μ_0 : permeability of free space, $4\pi \times 10^{-7}$ H/m

l : length of regular polygon electrode

d : width of regular polygon electrode

Supplementary Table 4 Parameter of regular polygon electrode

n	3	4	5	6	8
$f(n)$	-1.40546	-0.77401	-0.40914	-0.16152	0.21198

Supplementary Figure 10 | Theoretical inductance calculation of various shape electrodes. (a) Inductance of Ag NFs electrodes with different numbers of edges. (b) Comparison of inductance changes of different electrode lengths with the same area variation. (c) Inductance variation of circular electrodes with different radii.

As shown in Supplementary Figure 10a-c, for the electrodes with the same length but different shapes, the circular electrode exhibits the highest inductance due to its larger area. Meanwhile, Supplementary Figure 10b shows that the greater of the electrode length have the smaller inductance

change at the same area variation ratio. The results indicate that L is mainly affected by the dimension and layout of the coil, especially the electrode length. Supplementary Figure 10c illustrates that the inductance variation versus the circular electrode radii, which shows the same trend as the experimental results.

iii) Coupling effect of multi-turns coil electrodes

For the planar spiral coil electrode, the specific formula could be obtained by fitting as follows:

$$L = \frac{\mu_0}{8\pi} \omega^2 d \psi \quad (3-7)$$

μ_0 : permeability of free space, $4\pi \times 10^{-7}$ H/m

ω : number of turns of the coil

d : average diameter of the coil

ψ : a specific parameter which was determined by the cross section of electrode

It could be found that the spiral coil inductance shows a square proportional relationship with turns, resulting from the double number of inductive coupling between the internal coil for each additional coil turn.

(b) Modeling analysis

Supplementary Figure 11 | Research approach and the article logic structure.

As shown in Supplementary Figure 11, the quality factor is affected by the inductance and resistance. The resistance is strongly influenced by the dimension and material characteristics of the conducting electrodes, such as the length or width, NFs density and sputtering duration, which smaller resistance can be achieved by optimized synthesis process. Furthermore, we further analyze and fit the data to better control the Ag NFs electrode inductance (sheet resistance is $11.1 \sim 1.68 \Omega/\square$, at $>70\%$ transmittance).

The straight Ag NFs electrodes are first investigated as the most basic structural unit, and the empirical formula of the straight electrode length and width relative to the inductance could be obtained:

$$L = \frac{\mu_0 l}{2\pi} [\ln(l/d) - 0.912] \quad (3-8)$$

It could be further calculated to obtain the following formula:

$$L_{straight} \approx 2 \times 10^{-7} \times l \times [\ln(l/d) - 0.912] \quad (3-9)$$

Then, the inductance of the coil reduces from a straight shape to a shapely pattern due to the existence of mutual inductance between different line segments. The inductance of curve electrode can be described as:

$$L = L_{straight} - \Delta L \quad (3-10)$$

For a certain length electrode, the circular electrode has the largest area. So, we use the circular electrode as a reference object to explore the inductance variation with different areas. The effect of area on the inductance could be obtained by derivation of the previous formula:

$$\Delta L = \frac{L_{circle}}{S_{circle}} \Delta S + A \quad (3-11)$$

As shown in Figure 1h, the changes in inductance are linearly related to the radius of the coils. It can be fitted:

$$L \approx 0.966 L_{straight} - 1.867 \times 10^{-9} \quad (3-12)$$

$$\frac{L_{circle}}{S_{circle}} = \frac{19.095 \times 10^{-9}}{1.2732 \times 10^{-4}} = 14.998 \times 10^{-5} \quad (3-13)$$

Hence, the formula (3-10) can be modified as:

$$L = L_{straight} - \frac{L_{circle}}{S_{circle}} \Delta S - A = L_{straight} - 14.998 \times 10^{-5} \times \Delta S - A \quad (3-14)$$

After fitting and calculating the inductance of different shapes, empirical formula could be obtained:

$$L \approx L_{straight} - 1.4998 \times 10^{-4} \times \Delta S - 5.203 \times 10^{-10} \quad (3-15)$$

Where $L_{straight} = 2 \times 10^{-7} \times l \times [\ln(l/d) - 0.912]$. Supplementary Table 5 shows the error between the theoretical and the measured value.

Supplementary Table 5 Parameter of regular polygon electrode

Edge	3	4	5	6	7	8
L_{mea} (nH)	13.368	15.932	16.774	17.287	17.688	18.241
L_{cal} (nH)	12.066	14.027	15.493	16.277	16.645	17.038
Error	0.097	0.119	0.076	0.058	0.059	0.065

2. In Fig.2, the SEM images appear to have been taken in different spots, so it is difficult to compare cracks caused by deformation and the intrinsic structural gap after fabrication.

Response:

Thanks the reviewer for the suggestion. A real-time in-situ observation is much better, as the reviewer suggested.

(i) We added more detailed information in **Supplementary Note 1** to analysis the process of crack initiation and propagation, including using more samples and observing the microstructure of the Ag NFs in-situ. More observations of samples under different tensile strain are shown in **Supplementary Figure 3**.

(ii) As the reviewer suggested, **the SEM images of the sample in the same place are illustrated in Supplementary Figure 4** by using an array pattern marker. It's clear that the Ag NFs electrodes in the original state are smooth. At small extensional strains, some Ag NFs begin to break. And cracks generated with the strain sequentially increased, which do not be found in original state. So, these cracks are caused by the deformation of the electrode.

The Supplementary Note is revised according to your comment, which is listed as following for your convenience:

Supplementary Figures, Supplementary Note 1, Page S6-S7

Supplementary Note 1: Microstructure of Ag NFs under tensile strain

(a) SEM images of Ag NFs with different strains

The SEM images of Ag NFs electrodes with no stress are shown in the Supplementary Figure 2. The Ag NFs electrodes in the original state are randomly stacked and their surface is very smooth. The intrinsic gap of Ag NFs provides the good optical transparency of the electrode.

Supplementary Figure 2 | SEM images of Ag NFs electrodes in the original state without tensile strain. (a) SEM images of Ag NFs (sample 1) under different magnifications. (b) SEM images of Ag NFs (sample 2) under different magnifications.

Fractures or cracks will generate on the Ag NFs electrodes with tensile strain increased. The morphologies of four samples under different strains are shown in the Supplementary Figure 3, which are at the same preparation processing conditions. These results show that the fracture of Ag

NFs surface is rough and it has obvious difference from the intrinsic Ag NFs gap. On the other hand, the SEM images also depict that there are still effective conductive paths under tensile strain for the random orientation Ag NFs. These cracks will further expand with the increasing deformation, leading to the increment of the parasitic capacitance and the reduction of its inductance. In particular, the electrode loses its conductivity and will be completely transformed into a capacitor when the deformation exceeds the range that the Ag NFs can be withstand.

Moreover, the SEM in-situ observation are employed to further analysis the process of crack initiation and propagation by using an array pattern marker, as shown in the Supplementary Figure 4. It also indicates that the cracks expand with the increasing strain, in keeping with the previous discussed.

Supplementary Figure 3 | SEM images of Ag NFs under different tensile strain (without PDMS packaging). Four samples with the same processing conditions. (a-d) SEM images of Ag NFs under 15% strain (a, sample 1), 30% strain (b, sample 2), 45% strain (c, sample 3) and 60% strain (d, sample 4) at different magnifications.

Supplementary Figure 4 | SEM images of Ag NFs under different tensile strain (without PDMS packaging). All SEM images are taken at the same place. (a-e) SEM images of Ag NFs under 0% strain (a), 15% strain (b), 45% strain (c), 60% strain (d) and 100% strain (e) at different magnifications.

3. All deformation is limited under $\epsilon = 100\%$. More details should be included about the yield point to indicate that the $\epsilon = 100\%$ is an elastic region.

Response:

Thanks the reviewer for the suggestion.

In fact, there are two reasons to choose $\epsilon = 100\%$ as a reference standard:

(i) Biological skin is stretchable up to 75% on average, as shown in Table R1 (*Clinics in dermatology*, 1995, 13(4), 375-380; *Nature*, 2011, 474, 64-67). So, the Ag NFs electrodes will be able to work normally on the human skin if they can withstand 100% tensile strain.

Table R1 Strain of the skin in various parts of the body

Body parts	Stain
knuckles	80%
elbow joint	60%
knee	49%

(ii) The PDMS substrate used in our experiments determines the tensile properties of the whole PDMS/Ag NFs composite electrodes. For this type of the substrate, the strain is around 140% before breaking. As the reviewer suggested, **the mechanical behavior of the PDMS is studied** in this work by one-way drawing. A true stress-strain curves is obtained, indicating that the $\epsilon = 100\%$ is in the elastic region, detailed can be seen in **Supplementary Note 1**.

The Supplementary Note is revised according to your comment, which is listed as following for your convenience:

Supplementary Figures, Supplementary Note 1, Page S7-S8

Supplementary Note 1: Microstructure of Ag NFs under tensile strain

(b) Mechanical property of the PDMS substrate

The main agents and curing agents of the PDMS (SYLGARD184 Dow Corning) are mixed thoroughly in a weight ratio of 10:1 and then degassed for 5 min to remove air bubbles. Then, spinning it on the glass substrate and drying in the oven at 120 °C for 20 min. Supplementary Figure 5 shows the uniaxial tensile tests to evaluate the mechanical properties of the PDMS, which shows an ultimate stress of 2.238 MPa at the strain of 144%. And it could be found that PDMS is in the elastic region within the 100% deformation which could satisfy the human body skin deformation.

Supplementary Figure 5 | One-way drawing of the PDMS substrate. (a) Uniaxial tensile test of the PDMS substrate. (b) Schematic diagram of the measuring device.

4. From an electromagnetism viewpoint, the geometrical characteristics of circular coil's self-inductance is already very well known. So, Fig.3, 4, and 5 appear to be a validation of coil design rather than giving a new finding.

Response:

Thanks the reviewer for the concern.

This work reports highly stretchable transparent wireless electronics composing with Ag NFs spiral coils and other functional electronic components for the power transfer and information communication. The innovation of this work is described as follow:

- (i) A novel method to fabricate complex multilayer stretchable transparent electronics: High-precision patterned Ag NF electrodes in an elastomeric PDMS matrix could be designed via electrospinning, magnetron sputtering, photolithography, and wet etching, with the advantages of simple operation, low cost, and easily large-scale preparation which could be promising application in circuit systems (**Figure 1**). And this patterned preparation process is also very suitable for Au NFs, Pt NFs and Ag NFs and Cu NFs (**Supplementary Figure 6**)
- (ii) Systematic study on the wireless power transfer efficiency of the Ag NFs coils under tensile strain: RF properties of different Ag NFs coils are investigated via one-port scattering analysis method (**Figure 2-4, Supplementary Note 2-5**). Various cracks will generate in the Ag NFs electrodes when the device under strain, resulting in the change in the R_S and L of the coils. Hence, it's important to analyze the effects of the strain on the RF characteristic of the coil, and an optimized coil structure should be designed, considering the density of the Ag NFs, the shape of the electrode, and the coupling effect of multi-turn coils (**Figure R1a-b**). Indeed, from an electromagnetism viewpoint, the geometrical characteristics of circular coil's self-inductance is already very well known. However, for stretchable electronics, **the effects of the strain on performance and geometry of the coil is also important (Figure R1 c)**.

Fig. 3a: Effects of the density of the Ag NFs

Fig. 3c-d: Effects of the coupling effect and the strain

Fig. 4d: Effects of the strain

Figure R1 | Effects of the strain on performance of the coil.

- (iii) Wireless multifunction electronics for wearable devices: Various kinds of wireless power transfer and signal detection systems are developed by integrating the Ag NFs coil with other tiny electronic components, which could still work in the water or irregular plane. We have analyzed the wireless power transfer efficiency of the Ag NFs coil based on the magnetic resonance coupling transmission mode, and there are few reports about the RF properties of stretchable electrodes (**Figure 5, Supplementary Note 6**). Moreover, we have realized the short-distance content recognition by using an NFC tag and the long-distance audio transmission by employing FM technology, predicting that the present device has application in information identification systems, soft robotics and wearable electronics (**Figure 6, Supplementary Note 7**).

5. Other important parameters for AC resistance of conductors including permeability and conductivity of the medium is never mentioned. Please specify permeability and conductivity of the medium.

Response:

Thanks the reviewer for the suggestion.

In this work, the Ag NFs electrodes are packaged with PDMS which is a good insulator. As the reviewer suggested, **we have measured and calculated the magnetic permeability and electric conductivity of PDMS**, as shown in the **Supplementary Table 2**.

By using this fabrication process, we also have fabricated patterned Au NFs, Pt NFs, Ag NFs and Cu NFs electrode (**Supplementary Figure 6**). But, for chromium (Cr), nickel (Ni), it is difficult to obtain a good stretchable electrode due to their low ductility. Considering the cost and electrical properties among Au NFs, Pt NFs, Ag NFs and Cu NFs, Ag NFs is an ideal material used for the fabrication of the patterned stretchable electrode, detailed information can be seen in **Supplementary Note 1**.

The Supplementary Note is revised according to your comment, which is listed as following for your convenience:

Supplementary Figures, Supplementary Note 1, Page S8-S9

Supplementary Note 1: Microstructure of Ag NFs under tensile strain

(c) Conductivity and permeability of various materials

The magnetic permeability and electric conductivity of the materials are also important parameters for the device performance, including PDMS and metal nanofibers, as shown in Supplementary Table 2.

Supplementary Table 2 Conductivity and permeability of PDMS and metals

	σ (S/m)	μ_r
PDMS	3.45×10^{-13}	1
Ag	5.00×10^7	0.999974
Au	4.88×10^7	0.999983
Pt	9.52×10^6	1.000260
Cu	5.88×10^7	0.999900
Cr	7.92×10^3	1.000140
Ni	1.44×10^4	120

In this work, we also have fabricated a variety of metal nanofibers by using this method, including silver (Ag), gold (Au), platinum (Pt), copper (Cu), chromium (Cr), nickel (Ni). It could be noticed that this patterned preparation process is very suitable for Au NFs, Pt NFs and Ag NFs, Cu NFs, because these metals perform excellent ductility. However, the Cr NFs and Ni NFs will rupture on the surface of water, which make it impossible to form a hard film. Supplementary Figure 6 shows the patterned electrodes of Au NFs and Cu NFs. (Anderson, J. Malleability and Ductility of Metals. SCI AM, 1869, 21, 41-341)

Supplementary Figure 6 | Various patterned electrodes of nanofibers. (a) Optical image of the Au NFs. (b-c) Patterned Au NFs (b) and Cu NFs (c) electrode.

6. Authors discuss the stretching capabilities of the presented Ag NF coil, but a mechanical analysis has not been shown. It would be useful to model the coil to validate how the mechanical stress impacts its structure and if there are some critical/weak points. Moreover, for a wearable device, the mechanical stress is related to the body part where the device is applied itself, hence a crucial role is played by the design of the coil. In order to keep the antenna parameters as robust as possible while undergoing deformations (Fig. 3 shows how the deformation affects the inductance and the resistance) an example of ad-hoc design for a specific human body part application should be shown (e.g. the tensile stretch caused by the neck movement will be different from the one coming from an arm, as shown by C. Miozzi et al in “Radio-mechanical Characterization of Epidermal Antennas during Human Gestures”).

Response:

Thanks the reviewer for the suggestion. We have analyzed the mechanical stress of coil under tensile strain and added this information in the revised MS (**Figure 4**). A video has been added to observe the state of the device with elbow bending as well (**Supplementary Movie 1**).

(i) It is essential to analyze the mechanical stress of coil under tensile strain for better understanding the stress distribution of Ag NFs and corresponding improvements to satisfy the practical applications. Therefore, we have added this information in the revised MS (**Figure 4**). Simulation about unidirectional, bidirectional and random orientation Ag NFs under tensile strain are carried out to exhibit the corresponding stress distribution by using a finite-element method (COMOSL) in three-dimensional space (detailed can be seen in **Supplementary Note 5**). It was found that square electrodes could easily produce stress concentration at the corners, resulting in the break of the electrode, which is similar to the simulation results.

(ii) We have read relevant literatures about the deformation of the body part, including the reviewer mentioned paper. Biological skin is stretchable to 75% on average, as shown in Table R1 (*Clinics in dermatology*, 1995, 13(4), 375-380; *Nature*, 2011, 474, 64-67). Hence, the circular electrodes based on the random orientation Ag NFs could completely adhere to different parts of human body. Additionally, it's still a technical problem to detect the movement of different parts of human body in real-time. In the revised MS, a video has been added to observe the state of the device with elbow bending (**Supplementary Movie 1**).

The main text and the Supplementary Note are revised according to your comment, which is listed as following for your convenience:

Main text, Page 12

To further understand the variation of Ag NFs network under tensile strain, Ag NFs with different orientations (unidirectional, bidirectional, random) were simulated via finite element analysis method, as shown in Fig. 4a. The orientation of Ag NFs has a major impact on the stretchable property of the electrode. For the unidirectional Ag NFs network, the nanofibers have a significant stress concentration when the orientation of Ag NFs is in the similar direction of the tensile strain, especially at the interconnected junction. Similarly, the nanofibers along the tensile direction are still in a high-stress state for the bidirectional Ag NFs, while it is relatively lower in another direction. It is worth mentioning that the Ag NFs network with random structure has the lowest stress concentration, which means it is more likely to withstand higher strain. The corresponding SEM

images of these Ag NFs networks under strain are shown in the Fig. 4b (more analysis could be seen Supplementary Fig. 17 and Supplementary Note 5), indicating that the multi-orientation nanofibers possess various conductive paths even under high strain. Additionally, Figure 4c illustrates the stress distribution of different patterned Ag NFs electrodes (square, hexagonal and circular) under deformation. It is found that the electrode edges along the stretching direction have greater stress. And it is easier to generate the stress concentration at the corners of the patterned electrode where the large cracks are also more likely to develop with the increasing strain, as shown in Fig. 4d (detailed information can be seen in Supplementary Fig. 18-21). Therefore, circular electrodes based on the random orientation Ag NFs will perform the better tensile property and it's an ideal structural design for stretchable electronics.

Main text, Figure 4, Page 35

Fig. 4 Simulations of the mechanical properties of various Ag NFs networks. a Stress distribution of Ag NFs networks under tensile strain with different orientations (unidirectional, bidirectional, random). b SEM images of the Ag NFs networks under tensile strain (Top: bidirectional NFs, bottom: random NFs). The NFs break under stress, but there are still conductive paths in the electrodes. c Stress distribution of different patterned Ag NFs electrodes (square, hexagonal and circular) under strain. Stress concentration tends to occur in the corners of the electrodes. d Optical images of large cracks at the corners of square electrodes with the increasing strain (Top to bottom: 15%, 45% and 60%).

Supplementary Methods, Page S5

Finite element method (FEM) modelling.

The stretching mode is calculated by using the Solid mechanics modulus in COMSOL Multiphysics, and this model involves in the Ag NFs electrodes covered with PDMS. The PDMS substrate is fixed at one end while applying displacement on the other end. The Von Mises Stress yield criterion is applied to observe the stress distribution of Ag NFs. The wireless power transfer mode is calculated by using the radio frequency modulus. The mode consists of two Ag NFs coil encapsulated with PDMS, which is settled in an air domain with perfectly matched layers (PML). The surrounding PMLs are necessary to absorb the radiation from the transmitting antenna and analyze the antenna coupling in the infinite free space. A lumped port with 50 Ω reference impedance is assigned to excite or terminate the antennas.

Supplementary Note 5: Finite element simulation of the mechanical property

(a) Mechanical properties of nanofibers

Simulations of unidirectional, bidirectional and random orientation Ag NFs under tensile strain are carried out to exhibit the stress distribution by using a finite-element method (COMOSL) in three-dimensional space, as shown in Supplementary Figure 17a-c. It is worth noting that the Ag NFs have significant stress concentration when the deformation along the nanofiber's orientation, especially on the interconnection of the Ag NFs. Hence, unidirectional Ag NFs are susceptible to stress concentration which are not suitable for the stretchable devices.

For the bidirectional and random orientation Ag NFs with tensile strain, although the Ag NFs in one direction are in a state of high strain, Ag NFs in the other directions could maintain low stress which can stay intact to form a conductive path. The Supplementary Figure 17d shows the evolution of unidirectional Ag NFs under tensile strain. Upon stretching, the PDMS substrate with low effective stiffness presents elastic deformation to absorb stress and many fractures will generate on the Ag NFs along the tensile direction. Hence the Ag NFs electrode will change from an inductance to capacitance and the corresponding SEM (right) images illustrate the variation of the unidirectional Ag NFs under tensile strain.

The tensile properties of the bidirectional Ag NFs and more SEM images are shown in the Supplementary Figure 17e. It could be found that many fractures appear on the Ag NFs along the tensile direction, which is consistent with the simulation results. Nevertheless, Ag NFs along the other orientations are still kept intact and there exist various conductive paths in high strain. Moreover, random orientation Ag NFs have more orientations than that of bidirectional Ag NFs and it possesses more conductive paths under tensile strain. In addition, the SEM images also show the as-synthesized Ag NFs are naturally interconnected at their junctions. Hence, we speculate the random orientation Ag NFs perform the better tensile properties, which is an ideal material for epidermal electronics (even in the places with large strains of human body, including neck or joint, etc.).

(b) Mechanical properties of Ag NFs coils

Based on the above analysis, it could be found that Ag NFs with random orientation still exist various conductive paths even under high strain, so we equate the Ag NFs to a thin film to explore the optimal stress distribution under different shapes and design the better geometrical structure for stretchable devices. Detailed simulation results are shown below.

Simulations of stress distribution for different straight electrodes are shown in Supplementary Figure 18. We simplified the Ag NFs electrodes into a thin film in order to facilitate the calculation of the model. The Ag NFs electrodes with different lengths and widths are subjected to tensile analysis. The results show that the longer of the electrode length means intense stress concentration in the middle of the electrode, and the Ag NFs is prone to fracture. However, for the Ag NFs electrode with different widths, the wider electrode means less stress distribution in the middle. To quantitatively analyze the stress variation of Ag NFs electrode, we select the middle point for stress calculation, the results show that the stress have increased from 0.176 MPa to 0.367 MPa with the increase of the length, but declined from 0.263 MPa to 0.259 MPa with the increase of the width.

Supplementary Figure 17 | Simulation and SEM images of the Ag NFs with tensile strain. (a-c) Simulations of the stress distribution of unidirectional Ag NFs (a), bidirectional Ag NFs (b) and multi-orientation Ag NFs (c) under tensile strain. (d-e) Schematic diagram and SEM images of the evolution of the unidirectional Ag NFs (d) and multi-orientation Ag NFs (e) under tensile strain.

Supplementary Figure 18 | Finite element simulation of straight Ag NFs electrodes versus tensile strain. (a) Ag NFs electrodes with different lengths under tensile strain. (b) Ag NFs electrodes with different widths under tensile strain.

For the Ag NFs electrodes with different shapes (including triangle, square, pentagon, hexagon, octagon and circular), the simulation results are similar to that of the nanofibers. The polygon electrode edges with the same orientation as deformation are easier to bear the larger stress which

means the Ag NFs electrode are more likely to break. Moreover, the results also exhibit that the stress is easily to aggregate in the corner of the polygon electrode, which means the Ag NFs electrodes here are prone to fracture. Hence, it is better to choose the circular electrode as the stretchable devices so as to avoid the stress concentration at the corners.

Supplementary Figure 19 | Finite element simulation of Ag NFs electrodes with different shapes versus tensile strain.

Furthermore, we compared the stress distribution of circular electrodes with different lengths and widths. For the circular electrode, the electrode length does not affect the stress distribution excessively, but the wider electrode means the smaller stress at the boundary, which is more conducive to prepare stretchable devices. Therefore, the electrode width could be appropriately increased to obtain better tensile properties when designing the circular electrode.

Supplementary Figure 20 | Finite element simulation of circular Ag NFs electrodes with different lengths (a) or widths (b) under tensile strain.

Finally, we simulated the stress distribution of the Ag NFs electrodes (including quadrangle and circular) with different tensile directions. The schematic diagram of COMSOL model is shown in Supplementary Figure 21a, and the Ag NFs electrodes are encapsulated with PDMS and the bottom end of the PDMS is fixed. Subsequently, stress is applied at the two upper corners of the PDMS substrate, and the results in Supplementary Figure 21b shows that the stress distribution of square electrodes is obviously greater than the circular electrodes, especially at the corners.

Supplementary Figure 21 | Finite element simulation of Ag NFs electrodes with different tensile direction. (a) Schematic diagram of stretching model in COMSOL. (b) Simulation results of the square and circular electrodes.

7. In the last section, authors show a possible application of the antenna involving Frequency Modulated (FM) communication. In the mentioned test, there is lack of information about the carrier frequency. In this scenario, it would be interesting to provide a measurement of the IEEE Gain of the antenna, with an estimation of the maximum reading distance with a 4 W EIRP condition.

Response:

Thanks the reviewer for the suggestion. The carrier frequency range used in this work is 87 MHz to 108 MHz, and we choose 100 MHz as carrier wave to transmit audio information. As the reviewer suggested, we simulated the Ag NFs antenna's gain and estimation the maximum reading distance with a 4 W EIRP condition which can be seen in the **Supplementary Note 7**.

The Supplementary Note is revised according to your comment, which is listed as following for your convenience:

Supplementary Figures, Supplementary Note 7, Page S35-S36

Supplementary Note 7: Applications in functional wireless electronics

(c) Ag NFs antenna gain radiation pattern

The carrier frequency used in this work is 87 MHz to 108 MHz, and we choose 100 MHz as carrier wave to transmit audio information. In order to deeply understand the RF performance of Ag NFs antenna, we simulated the gain of the antenna via finite-element method (COMSOL) in three-dimensional space. The reference model is simulated with the same dimension as that of Ag NFs antenna and it is enclosed by a numerical version of a sphere which is a perfect matched layer (PML). The Ag NFs antenna is encapsulated with PDMS, and the lumped port has 50 Ω for reference impedance and all domains is filled with air. Supplementary Figure 30a shows that the Ag NFs spiral coil antenna gain at 0.1 GHz, and it could be seen that the antenna's gain is -15.48 dBi. Subsequently, the most important tag performance is the read range - the maximum distance at which RFID reader can detect the backscattered signal from the tag. Therefore, we further analyzed the antenna's gain at 0.9 GHz which is the UHF bands for RFID systems, and it could be seen that the antenna's gain is -0.16 dBi as shown in the Supplementary Figure 30b. According to the literature⁷, under the hypothesis of polarization matching between the reader and tag antennas, the maximum activation distance of the tag along the (θ, ϕ) direction is then given by Equation 7-2.

$$d_{\max}(\theta, \phi) = \frac{c}{4\pi f} \sqrt{\frac{EIRP}{P_{chip}} \tau G_{tag}(\theta, \phi)} \quad (7-2)$$

c : velocity of the light in free space;

f : working frequency of the antenna;

P_{chip} : sensitivity of the chip;

G_{tag} : gain of the antenna;

EIRP: the effective power transmitted by the reader;

The factor

$$\tau = \frac{4 R_{chip} R_A}{|Z_{chip} + Z_A|^2} \quad (7-3)$$

is the power transmission coefficient, which accounts for the impedance mismatch between the antenna ($Z_A = R_A + jX_A$) and the microchip ($Z_{chip} = R_{chip} + jX_{chip}$). The impedance of microchip

depends on the input power, and its input reactance is strongly capacitive since the transponder includes an energy-storage. The antenna impedance should be inductive in order to achieve conjugate matching, and a large impedance phase angle needs to be obtained. Beyond d_{\max} , the power collected by the tag decreases below the microchip sensitivity, and the tag becomes unreachable. The input impedance and sensitivity of the chip is $Z_{chip} = 12 - j150$ and $12\mu\text{W}$, the impedance of prepared Ag NFs antenna could be $Z_A = 98 + j150$ to achieve the impedance matching. Hence the power transmission coefficient could be calculated as 0.0433, and substitute τ into the equation 7-2 to estimate the maximum reading distance with a 4 W EIRP condition.

$$d = \frac{3 \times 10^8}{4\pi \times 9 \times 10^8} \sqrt{\frac{4}{12 \times 10^{-6}} \times 0.0433 \times 0.96} = 3.12 \text{ m} \quad (7-4)$$

Supplementary Figure 30 | Finite element simulation of Ag NFs antenna gain radiation pattern. (a) Simulated Ag NFs antenna gain patterns at 0.1 GHz. (b) Simulated Ag NFs antenna gain patterns at 0.9 GHz.

In addition, we also analyzed the Ag NFs antenna's gain variation under tensile strain and it could be noticed that the gain decreases from -0.16 dBi to -0.33 dBi. The maximum reading distance with a 4 W condition changes to the 3.05 m, which scarcely change compared to the original device, indicating that the Ag NFs antenna has extensive applications in wearable electronic devices.

8. Give specific descriptions instead of using vague wording - line 104, “excellent wireless power transfer efficiency”: it should be careful to say that 15% power transfer efficiency (PTE) is excellent without reference. Please add appropriate reference to clarify this sentence.

Line 336, “integrating the Ag NF coil with tiny electronic components”: please specify the size of components for clarity.

Response:

Thanks the reviewer for the suggestion. As the reviewer suggested, we modified the ambiguous words in the MS with a rigorous expression. We also summarized the pioneering works about transmission efficiency, as shown in **Supplementary Table 6**. The efficiency of commercial wireless power transmission systems based on the magnetic induction is up to 95%. However, few reports are available about stretchable antenna for power transfer with high transmission efficiency. In this work, the radio frequency characteristics of Ag NFs antenna was explored by designing the optimal geometry for better performance. In addition, we illustrated the specific dimensions of tiny electronics in **Supplementary Figure 27**, and we modified the ambiguous words in the MS with a rigorous expression.

The main text and the Supplementary Note are revised according to your comment, which is listed as following for your convenience:

Main text, Page 5

Moreover, the working frequency of the device should be set below the self-resonance frequency f_0 or as the frequency with the maximum value of Q for better performance. Based on the magnetic resonant coupling mode, we find that a five-turn stretchable spiral coil possesses 15% wireless power transfer efficiency with $\varepsilon = 100\%$. Furthermore, multifunction wireless electronics and signal detection systems are developed by integrating the Ag NFs spiral coils with other tiny electronic components.

Main text, Page 15

Wireless power transfer and signal detection system. The stretchable transparent Ag NFs spiral coil can be integrated with other tiny electronic components ($1.6 \text{ mm} \times 1.2 \text{ mm} \times 0.45 \text{ mm}$, Supplementary Fig. 27) and assembled into more complex functional wireless electronics for power transfer and data communication. Figure 6a shows a typical device structure that is easy to operate. PDMS was employed as the substrate, the insulator layer, and the encapsulation layer, which were transparent and stretchable to absorb the strain energy.

Supplementary Figures, Supplementary Note 7, Page S31-S32

Supplementary Note 7: Applications in functional wireless electronics

(a) Stretchable antenna for power transfer

The wireless power transfer efficiency is also an important parameter for the magnetic induction antenna. It has been reported that the transmission efficiency can reach 95%, and there is no obvious advantage from the perspective of power transfer efficiency for our devices and systems. However, few reports are available about stretchable antenna for power transfer with high transmission efficiency. In this work, the radio frequency characteristics of Ag NFs antenna is explored by designing the optimal geometry for better performance. The results show that a five-

turn spiral coil possesses 40% power transfer efficiency at 20 MHz, and the transfer efficiency was approximately 15% even under the strain of 100% based on the magnetic inductive coupling mode. Supplementary Table 6 shows the comparison of the current magnetic inductive transfer efficiency with this work, indicating that Ag NFs antenna has broad application in human epidermal electronics.

Supplementary Table 6 Summary of the pioneering works about transmission efficiency

Power	Frequency	Gap (mm)	Transmitter area (cm ²)	Receiver area (cm ²)	Materials	Efficiency η %	Efficiency with strain	References
0.1 mW	20 MHz	2	-5.72	-4.55	Ag NFs	-40%	-15%	This work
-	3 ~ 5GHz	1	-	-	Ag NWs	-	-	Nat. Commun. 2017, 14997
-	50MHz	5	-	-1.13	Ag NFs	21.5%	-	Sci. Adv. 2018, eaap9841
-	13.4MHz	-	-	-1.77	Cu NFs	-	-	Adv. Sci. 2018, 1801146
0.1 W	6.78 MHz	15	21	0.785	30 AWG wire	10 ~ 20%	-	ISCCAS, 2007, 2080-2083.
0.4 ~ 2 W	500 KHz	-	520	50	-	35%	-	EPE, 2009, 1-10
0.794 W	27 MHz	15	-	-4	-	80%	-	ISABEL, 2010, 1-5
1.2 W	500 KHz	130	314	314	Wire windings	40%	-	ECCE, 2013, 2239-2244
25.6 W	13.56 MHz	-	-28.3	-28.3	concentric coils	73.4 %	-	IEEE MTT-S International, 2012, 1-3.
50 W	3.54 MHz	300	1257	1257	Cu on the PCB layouts	80%	-	ECCE, 2013, 1917-1924
100 W	20 KHz	700	707	707	Litz wires AWG 36600	95.4%	-	IEEE MTT-S International, 2012, 83-86.
2 KW	5 ~ 50 KHz	50 ~ 80	1385-3849	1385-3849	Litz wires	85%	-	ECCE, 2009, 2081-2088
5 KW	20 KHz	246	21*103	21*103	AWG36 Litz wire	90%	-	IEEE Trans. Ind. Inform., 2012, 585-595

The specific dimensions of the chip LED used in this work are shown in the Supplementary Figure 27. The chip LED could achieve a good contact with Ag NFs spiral coil by using lithography and 3D printing technology. It will be lighted using the external coil coupling with the Ag NFs spiral coil, indicating that the Ag NFs could be combined with traditional small electronic devices to achieve more human-machine interaction.

Supplementary Figure 27 | Specific dimensions of the chip-LED.

Reviewer #2:

Zhang and coauthors demonstrated to fabricate High-precision epidermal radio frequency antenna based on nanofiber network for wireless stretchable multifunction electronics. Ag nanofiber (NF) is attracting much attention because it has very high conductance and high transparency, which cannot be achieved with conventional metals. I understand the importance of the Ag NF and it will open the new era of flexible and stretchable electronics.

However, I could not understand the novelty of this work. For example, authors described “Notably, the Ag NF electrodes have greater electrical resistance than commercial electrodes owing to their interconnected junctions.” But I could not understand why the Ag NF based electrodes has the electrical characteristics greater than commercial electrodes. Authors should carefully explain the mechanism of the interconnected junctions. This manuscript dose not describe the microstructure and electrical characteristics with stretching and bending. At least, authors should explain the following questions and contain the experimental data for showing the novelty of the work and for better understanding of the proposed materials and the antenna.

Answers:

We like to express our sincere thanks to the referee for her/his great effort to review the manuscript and positive evaluation on our work.

1. The micrographs of the antennas with stretching and/or bending should be contained for showing the changes in the nanofiber networks.

Response:

Thanks the reviewer for the suggestion.

As the reviewer suggested, the micrographs of the antennas with stretching and/or bending are added in the revised MS and **Supplementary Note**, including **micrographs of Ag NFs with different strains (Supplementary Note 1)** and finite element simulation of **the mechanical properties (Supplementary Note 5)**.

The Supplementary Note is revised according to your comment, which is listed as following for your convenience:

Supplementary Figures, Supplementary Note1, Page S6-S7

Supplementary Note 1: Microstructure of Ag NFs under tensile strain**(a) SEM images of Ag NFs with different strains**

The SEM images of Ag NFs electrodes with no stress are shown in the Supplementary Figure 2. The Ag NFs electrodes in the original state are randomly stacked and their surface is very smooth. The intrinsic gaps of Ag NFs show a good optical transparency of the electrode.

Supplementary Figure 2 | SEM images of Ag NFs electrodes in the original state without tensile strain. (a) SEM images of Ag NFs (sample 1) under different magnifications. (b) SEM images of Ag NFs (sample 2) under different magnifications.

Fractures or cracks will generate on the Ag NFs electrodes with tensile strain increased. The morphologies of four samples under different strains are shown in the Supplementary Figure 3, which are at the same preparation processing conditions. These results show that the fracture of Ag NFs surface is rough and it has obvious difference from the intrinsic Ag NFs gap. On the other hand, the SEM images also depict that there are still effective conductive paths under tensile strain for the random orientation Ag NFs. These cracks will further expand with the increasing deformation, leading to the increment of the parasitic capacitance and the reduction of its inductance. In particular, the electrode loses its conductivity and will be completely transformed into a capacitor when the deformation exceeds the range that the Ag NFs can be withstand.

Moreover, the SEM in-situ observation are employed to further analysis the process of crack initiation and propagation by using an array pattern marker, as shown in the Supplementary Figure 4. It also indicates that the cracks expand with the increasing strain, in keeping with the previous discussed.

Supplementary Figure 3 | SEM images of Ag NFs under different tensile strain (without PDMS packaging). Four samples with the same processing conditions. (a-d) SEM images of Ag NFs under 15% strain (a, sample 1), 30% strain (b, sample 2), 45% strain (c, sample 3) and 60% strain (d, sample 4) at different magnifications.

Supplementary Figure 4 | SEM images of Ag NFs under different tensile strain (without PDMS packaging). One sample at the same place. (a-e) SEM images of Ag NFs under 0% strain (a), 15% strain (b), 45% strain (c), 60% strain (d) and 100% strain (e) at different magnifications.

Supplementary Figures, Supplementary Note 5, Page S21-S22

Supplementary Note 5: Finite element simulation of the mechanical property

(a) Mechanical properties of nanofibers

Simulations of unidirectional, bidirectional and random orientation Ag NFs under tensile strain are carried out to exhibit the stress distribution by using a finite-element method (COMOSL) in three-dimensional space, as shown in Supplementary Figure 17a-c. It is worth noting that the Ag NFs have significant stress concentration when the deformation along the nanofiber's orientation, especially on the interconnection of the Ag NFs. Hence, unidirectional Ag NFs are susceptible to stress concentration which are not suitable for the stretchable devices.

For the bidirectional and random orientation Ag NFs with tensile strain, although the Ag NFs in one direction are in a state of high strain, Ag NFs in the other directions could maintain low stress which can stay intact to form a conductive path. The Supplementary Figure 17d shows the evolution of unidirectional Ag NFs under tensile strain. Upon stretching, the PDMS substrate with low effective stiffness presents elastic deformation to absorb stress and many fractures will generate on the Ag NFs along the tensile direction. Hence the Ag NFs electrode will change from an inductance to capacitance and the corresponding SEM (right) images illustrate the variation of the unidirectional Ag NFs under tensile strain.

The tensile properties of the bidirectional Ag NFs and more SEM images are shown in the Supplementary Figure 17e. It could be found that many fractures appear on the Ag NFs along the tensile direction, which is consistent with the simulation results. Nevertheless, Ag NFs along the other orientations are still kept intact and there exist various conductive paths in high strain. Moreover, random orientation Ag NFs have more orientations than that of bidirectional Ag NFs and it possesses more conductive paths under tensile strain. In addition, the SEM images also show the as-synthesized Ag NFs are naturally interconnected at their junctions. Hence, we speculate the random orientation Ag NFs perform the better tensile properties, which is an ideal material for epidermal electronics (even in the places with large strains of human body, including neck or joint, etc.).

Supplementary Figure 17 | Simulation and SEM images of the Ag NFs with tensile strain. (a-c) Simulations of the stress distribution of unidirectional Ag NFs (a), bidirectional Ag NFs (b) and random orientation Ag NFs (c) under tensile strain. (d-e) Schematic diagram and SEM images of the evolution of the unidirectional Ag NFs (d) and multi-orientation Ag NFs (e) under tensile strain.

2. Describe what is the new compared with conventional Ag nanowire-based electrodes. If interconnected junctions are superior to the conventional nanowires, please explain the reasons. For example, Supplementary Figure 5 shows the SEM images of Ag NFs under tensile strain, but does not contain the changes in the situation of junctions. For better understanding of the novelty, data of the junctions is indispensable.

Response:

Thanks the reviewer for the suggestion.

(1) As the reviewer suggested, we add a comparison about the difference between the as-synthesized Ag NFs and traditional Ag NWs, detailed can be seen in **Supplementary Table 1**. On the one hand, for our as-synthesized Ag NFs, it will have significant cross-linking at their junctions, because the silver is deposited on the surface of the PVA NFs, as shown in Figure R2. These Ag NFs are naturally

interconnected at their junctions during the metal deposition, and it will make the Ag NFs possess a high conductivity close to their value in bulk. On the other hand, the Ag NFs with random orientations possess better mechanical property under tensile strain. Some fractures will appear on the Ag NFs along the tensile direction, while there are still conductive paths in the electrodes for Ag NFs with various orientations, detailed can be found in **Supplementary Note 5**.

Figure R2 | Cross-sectional SEM images of the Ag NFs.

We also fabricate a variety of metal nanofibers using this method, including Au NFs, Pt NFs and Ag NFs, Cu NFs. It is found that this fabrication process is very suitable for patterned the metal of good ductility, detailed information can be seen in **Supplementary Note 1**.

- (2) The remarkable performance of our as-synthesized Ag NFs could be attributed three main factors:
- (i) Traditional Ag NWs usually have lower conductivity compared to the bulk materials due to the impurities incorporated during synthesis, leading to the reduction of the crystal quality, large junction resistance and electron scattering. In contrast, our as-synthesized Ag NFs exhibit a much higher electrical conductivity than traditional Ag NWs, resulting from the high aspect ratio and the high-quality. The length of one Ag NFs can be more than 10 cm, so the current can go along the straight Ag NFs.
 - (ii) A highly uniform, interconnected network can be formed by using Ag NFs with different orientation. It has significant cross-linking at their junctions, because the silver is deposited on the surface of the PVA NFs, as shown in the Figure R2. These Ag NFs are naturally interconnected at their junctions during the metal deposition, and it also avoids the creation of a large junction resistance.
 - (iii) The Ag NFs network possesses excellent tensile properties contributed from the percolation theory and more conductive paths. Although some fractures appeared on the Ag NFs along the tensile direction, there are still conductive paths in the random orientation Ag NFs electrodes due to the more junctions of Ag NFs.

The Supplementary Note is revised according to your comment, which is listed as following for your convenience:

Supplementary Table, Page S1-S2

Summary of the state-of-the-art several key works about the Ag NFs

Conventional Ag NWs electrodes: the conventional Ag NWs are grown by reducing the Ag NO₃ in the EG solution. The length of Ag NWs is 20 to 150 μm, and the diameter is 30 to 150 nm. The Ag NWs solution is uniformly distributed in the elastomer matrix, which could fully embrace the percolated conductive filler network and protect it from strain-induced fracture. Although the electrodes based on the conventional Ag NWs possess the excellent stretchability, more NWs junctions lead to a large resistance and it is difficult to make patterns by using photolithography.

Some research works have been reported that the NWs junction could be improved by flash-induced electron excitations or depositing other materials on the interconnection, such as Au, Pt, CNTs, etc. Despite these attempts, none of the alternatives have succeeded in demonstrating high performance of all transparency, conductivity, stretchability and high-precision patterned electrodes.

As-synthesized Ag NFs:

(1) PVA NFs prepared by electrospinning possess a high aspect ratio, and the length of nanofibers can reach up to several centimeters. This will improve the electrode transparency and conductivity.

(2) Magnetron sputtering is a simple method for preparation of Ag NFs, which will deposit silver on the surface of PVA NFs. Noteworthily, Ag NFs are naturally interconnected at their junctions during metal deposition, which will make the Ag NFs possess the higher conductivity than conventional Ag NWs. The thickness and diameter of the Ag NFs could be controlled by sputtering time and electrospinning parameters, respectively. Moreover, the Ag NFs transfer process is carried out on the water surface, which can not only ensure the Ag NFs are flat on the PDMS substrate but also promote the dissolution of PVA. Meanwhile, the adhesion force between Ag NFs and substrate becomes stronger owing to the Van der Waal's force after drying.

(3) The Ag NFs could be patterned by using the photolithography technique and wet etching, and the line width was exactly up to several tens of micrometers. The patterned Ag NFs will be packaged by liquid PDMS, which will help the electrodes with the excellent tensile properties. Furthermore, Ag NFs will form the core-shell structure after dissolving the PVA NFs which will weaken the skin effect to a certain extent, and it will be helpful for designing radio-frequency electronics.

Here, we have explored a set of mature technology for preparing various Ag NFs electrodes or radio frequency devices, and then do some comparisons for electrical and mechanical properties with other works. The results show that our as-synthesized Ag NFs electrodes exhibit the advantages of simple operation, low cost and easily large-scale preparation, which will provide a powerful platform for wearable electronics.

Supplementary Table 1 Summary of the state-of-the-art several works about the Ag NFs

Fabrication methods	Materials	Structure	Conductivity	Stretchability	Transmittance	Patterning accuracy	References
Electrospinning & Sputtering	Ag NFs/ PDMS	Core-shell Ag NFs embedded in PDMS	$2.78 \times 10^4 \text{ S cm}^{-1}$ & $1.68 \Omega \text{ sq}^{-1}$	100%	70%	20 μm	This work
Electrospinning & Homogeneous dispersion	Ag NWs/ (PU) NFs	Ag NWs randomly distributed PU NFs scaffold	9190 S cm^{-1}	310%	—	—	Adv. Mater. 2019 , 31 , 1903446
Electrospinning & facile vacuum filtration	Ag NWs/ PA6 NFs	Ag NWs embedded into scaffold-reinforced conductive nanonetwork	$8.2 \Omega \text{ sq}^{-1}$	—	84.9%	—	ACS Nano 2018 , 12 , 9326
Electrospinning & Sputtering	Ag NFs	Continuous nanotrough networks with concave structure	$1.5 \Omega \text{ sq}^{-1}$	50%	90%	—	Nat. Nanotech. 2013 , 8 , 421
Electrospinning & Sputtering	Au nanomesh	Au nanomesh attach on the skin	$2.9 \times 10^3 \text{ S}$	48%	—	500 μm	Nat. Nanotech. 2017 , 12 , 907
Electrospinning & Sputtering	Au NFs/ PAN	Core-shell Au NFs on the PDMS	$25 \Omega \text{ sq}^{-1}$	80%	82%	—	Adv. Mater. 2013 , 3 , 1332
Homogeneous dispersion	Ag NWs/ SBS	Ag NWs ($l \approx 30 \mu\text{m}$, $d \approx 150 \text{ nm}$)	$1.2 \times 10^4 \text{ S cm}^{-1}$	Conductivity maintained up to 100% strain	—	—	ACS Nano 2015 , 9 , 6626
Homogeneous dispersion	Ag-Au NW/SBS	Core-shell Ag-Au NFs embedded in SBS	$4.18 \times 10^4 \text{ S cm}^{-1}$	266%	—	500 μm	Nat. Nanotech. 2018 , 13 , 1048
Homogeneous dispersion & Screen Printing	Ag NW/methyl cellulose	Ag NWs ($l \approx 20 \mu\text{m}$, $d \approx 30 \text{ nm}$)	$4.67 \times 10^4 \text{ S cm}^{-1}$	70%	—	50 μm	Adv. Mater. 2016 , 28 , 5986

Vacuum assisted filtration and transfer	Ag NW/PUA	Multiple times grown long Ag NWs ($l \approx 150 \mu\text{m}$), pre-stained eco-flex	$9 \Omega \text{sq}^{-1}$	460%	—	—	Adv. Mater. 2012 , 24 , 3326
Ag NW network soaked in GO dispersion	Ag NW-rGO/ PUA	GO reinforced Ag NFs junction connection	$14 \Omega \text{sq}^{-1}$	140%	75%	—	ACS Nano 2014 , 8 , 1590
Homogeneous dispersion	Ag NW/ polyacrylate	Ag NWs embedded in the surface layer of polyacrylate matrix	$7.5 \Omega \text{sq}^{-1}$	50%	80%	—	Nanotechnology 2012 , 23 , 344002
Homogeneous dispersion	Ag NW/PDMS	Partially embedded in elastomer (casting and peeling off)	$2.64 \Omega \text{sq}^{-1}$	35%	62%	—	J. Mater. Chem. C 2014 , 2 , 10369
Homogeneous dispersion	Ag NW/PDMS	Partially embedded in elastomer (casting and peeling off)	8130 S cm^{-1}	80%	—	—	Adv. Mater. 2012 , 24 , 5117
Homogeneous dispersion	Ag NW/NIPAM	NIPAM polymerization on Ag NW aerogel	93 S cm^{-1}	800%	—	—	Nat. Commun. 2018 , 9 , 2786
Homogeneous dispersion	Ag NW- PEDOT:PSS/PU	Brush painting hybrid ink on PU substrate	$19.7 \Omega \text{sq}^{-1}$	30%	88%	—	Sci. Rep. 2017 , 7 , 14685
Homogeneous dispersion	Ag NW- PEDOT:PSS/ Diels-Alder elastomer	Lock Ag NW network with PEDOT	$1.5 \Omega \text{sq}^{-1}$	100%	78%	—	ACS Appl. Mater. Interfaces 2015 , 7 , 14140
Homogeneous dispersion	Ag NW-carbon nanofibers/PU	Ag NW-carbon nanofibers coated on the surface of PU foam	16.6 S cm^{-1}	140%	—	—	Adv. Mater. Technol. 2019 , 1900060
Homogeneous dispersion	Ag NWs	A periodic two-dimensional network	$6.5 \Omega \text{sq}^{-1}$	—	91%	—	Nano Lett. 2012 , 12 , 3138
Homogeneous dispersion	Ag NWs/ PVA	PVA solution spin-coated over Ag NWs network	$70 \Omega \text{sq}^{-1}$	—	88%	—	Adv. Mater. 2010 , 22 , 4484
Homogeneous dispersion	Ag NWs	Ag NWs ($l \approx 10 \mu\text{m}$) & coated Au on junction	$20 \Omega \text{sq}^{-1}$	—	80%	—	ACS Nano 2010 , 4 , 2955
Heterogeneous assembly	Ag NP/ SBS	Ag NP precursor absorbed in SBS fiber	5400 S cm^{-1}	100%	—	200 μm	Nat. Nanotech. 2012 , 7 , 803
Homogeneous dispersion	Ag NWs	Long Ag NWs ($l \approx 150 \mu\text{m}$, $d \approx 100 \text{ nm}$)	$15.6 \Omega \text{sq}^{-1}$	—	90%	—	Nano-Micro Lett. 2015 , 7 , 51
Homogeneous dispersion	Ag-Au core-shell NWs Pt black/ SBS	Free-standing composite film	11210 S cm^{-1}	50%	—	500 μm	Adv. Mater. Technol. 2019 , 00768
Direct printing	Ag NWs	Large-scale-aligned Ag NWs ($d \approx 695 \text{ nm}$)	$26.9 \Omega \text{sq}^{-1}$	—	94.7%	—	Adv. Mater. 2016 , 28 , 9109

Supplementary Figures, Supplementary Note 1, Page S8-S9

Supplementary Note 1: Microstructure of Ag NFs under tensile strain

(c) Conductivity and permeability of various materials

In this work, we also have fabricated a variety of metal nanofibers by using this method, including silver (Ag), gold (Au), platinum (Pt), copper (Cu), chromium (Cr), nickel (Ni). It could be noticed that this patterned preparation process is very suitable for Au NFs, Pt NFs and Ag NFs, Cu NFs, because these metals perform excellent ductility. However, the Cr NFs and Ni NFs will rupture on the surface of water, which make it impossible to form a hard film. Supplementary Figure 6 shows the patterned electrodes of Au NFs and Cu NFs. (Anderson, J. Malleability and Ductility of Metals. SCI AM, 1869, 21, 41-341)

Supplementary Figure 6 | Various patterned electrodes of nanofibers. (a) Optical image of the Au NFs. (b-c) Patterned Au NFs (b) and Cu NFs (c) electrode.

3. The demonstration movies are not new and does not contain the scientific novelty of the work. No stretching and bending in the movies.

Response:

Thanks the reviewer for the suggestion. We have revised the **Supplementary movie 1** and added the performance of Ag NFs antenna under stretching and bending in the movie.

In summary, although Ag NF is one of the most interesting materials toward new flexible and stretchable electronics, this paper only shows the characteristics of an antenna made of nanowires, but does not give any information on the nanostructure behind it or any structural characteristics when the coil is deformed. Authors should carefully compare their work with pioneering works related to Ag nanowires and then describe the novelty of the Ag NF based antennas. Comparison table related to nanofiber-based antenna may be effective.

Response:

Thanks the reviewer for the concern and suggestions. We have made the corresponding revisions in the revised MS, including:

- (1) We have added corresponding experiments and simulations on the nanostructure of Ag NFs and the structural characteristics when the coil is deformed in the revised MS (**Supplementary Note 1, Supplementary Note 5**).
- (2) We have carefully compared our work with pioneering works related to Ag nanowires and describe the novelty of the Ag NFs based antennas. A comparison table is added in the **Supplementary section (Supplementary Table 1)**. The Ag NFs fabricated via electrospinning and magnetron sputtering have good electrical performance. For instance, the Ag NFs network interconnection junction prepared by electrospinning and magnetron sputtering have already been connected which have excellent conductivity than conventional Ag NWs. And, our as-synthesized Ag NFs network possess the good transparency and stretchability.

In Summary, this work reports highly stretchable transparent wireless electronics composing with Ag NFs spiral coils and other functional electronic components for the power transfer and information communication. The scientific and technological innovations in this work are as follows:

- i. A novel method to fabricate complex multilayer stretchable transparent electrodes: High-precision patterned Ag NFs electrodes in an elastomeric PDMS matrix could be designed via electrospinning, magnetron sputtering, photolithography, and wet etching, with the advantages of simple operation, low cost, and easily large-scale preparation which could be promising application in circuit systems. And this patterned preparation process is also very suitable for Au NFs, Pt NFs and Ag NFs and Cu NFs (**Supplementary Figure 6**)
- ii. Systematic study on the wireless power transfer efficiency of Ag NFs coils under tensile strain: RF properties of different Ag NFs coils are investigated via one-port scattering analysis method. Various cracks will generate in the Ag NFs electrodes when the device under strain, resulting in the change in the R_S and L of the coils. The L could be significantly enhanced to satisfy the practical applications, and the working frequency of the device should be chosen before the self-resonance frequency f_0 owing to the effect of strain. Besides, we analyze the wireless transmission circuit and find the coil exhibits outstanding wireless transmission capability even in the tensile state, which is approximately 15% in 10 MHz for five-turn stretchable spiral coil with $\varepsilon = 100\%$.
- iii. Wireless multifunction electronics for wearable devices: Various kinds of wireless power transfer and signal detection systems are developed by integrating the Ag NFs coil with other tiny electronic components which could still work in the water or irregular plane. Furthermore, we have realized the short-distance content recognition by using an NFC tag and the long-distance audio transmission by employing FM technology, predicting that the present device has application in information identification systems, soft robotics and wearable electronics.

Reviewer #3:

This manuscript entitled ‘High-precision epidermal radio frequency antenna based on nanofiber network for wireless stretchable multifunction electronics’ reports fabrication of flexible, wireless devices using Ag nanofiber spiral coils. The nanofibers are processed through lithography and wet etching to create several patterns. The authors have carried out significant investigation on the properties of the coil, including Quality factor and Radio frequency by considering the effects of strain on resistance and inductance of the coils. The novel synthesis technique combining electrospinning, lithography and wet-etching empowers the authors to synthesize precise electrodes which can be investigated for crack induction and its consequent effects on transmission capabilities. Overall, this manuscript is well written and experiment process is neatly organized. Especially, by analyzing the different shapes, turns and length of the coils which are important factor for influencing the magnetic field, and consequently the wireless transmission capability via the magnetic resonant coupling mode, the authors present a holistic evaluation of the coils and the synthesized devices. The reviewer thinks that this manuscript could give insights for the future of precision synthesis of conductive electrodes, and inspire efficient combinations of techniques to advance flexible electronics. Therefore, as a reviewer, I recommend that this manuscript can be published in Nature Communications after **Minor revision**.

Answers:

We like to express our sincere thanks to the referee for her/his great effort to review the manuscript and positive evaluation on our work.

1. In the Introduction part, additional references related to the magnetic resonance coupling transmission mode should be added to give more background to the reader.

Response:

Thanks the reviewer for the suggestion. More references about the magnetic resonance coupling transmission mode have been added to the revised manuscript.

The References is revised according to your comment, which is listed as following for your convenience:

References

40. Assawaworrarit, S., Yu, X. F. & Fan, S. H. Robust wireless power transfer using a nonlinear parity-time-symmetric circuit. *Nature* **546**, 387 (2017).
41. Imura, T. & Hori, Y. Unified Theory of Electromagnetic Induction and Magnetic Resonant Coupling. *Electr. Eng. Jpn.* **199**, 58-80 (2017).
42. Sample, A. P., Meyer, D. A. & Smith, J. R. Analysis, Experimental Results, and Range Adaptation of Magnetically Coupled Resonators for Wireless Power Transfer. *IEEE Trans. Ind. Electron.* **58**, 544-554 (2011).

2. Since the durability of these flexible devices is an important aspect of the whole synthesis process, a measure for the adhesion of the nanofibers on the substrate would be a valuable addition to the manuscript.

Response:

Thanks the reviewer for the suggestion.

(1) The Ag NFs are adhered to the PDMS substrate owing to the Van der Waal's force while they are being dried, detailed can be found in **Supplementary Note 1**. The surface of the Ag NFs film will become hydrophobic when the Ag NFs were firmly bonded to the substrate, so it is important to make the nanofibers hydrophobic. The Ag NFs film was dried by using the air gun while transferring the Ag NFs on the PDMS substrate, then immerse the sample into the water and blow dry again. After repeated the above process several times, fully dry the sample at 100 °C for more than 5 minutes to make the surface of the Ag NFs hydrophobic. Through the above simple methods, the surface of the Ag NFs could be modified. As shown in the Figure R3a and R3b, the contact angle of Ag NFs on the PDMS substrate with air gun blow dry was 58.60°. After fully drying the Ag NFs at 100 °C, it could be found that the contact angle becomes 127.73° which means the Ag NFs surface is hydrophobic, indicating that the water molecules in it have been completely evaporated and Ag NFs adhere well to the PDMS substrate.

(2) The fabricated Ag NFs electrodes will be packaged by another PDMS layer and the Ag NFs will not exfoliate from the PDMS substrate. The Figure R3c shows excellent stability by repetitive tensile testing over more than 3000 cycle, indicating that the device possesses the good stability.

Figure R3. The contact angle of Ag NFs on the PDMS substrate.

The Supplementary Note is revised according to your comment, which is listed as following for your convenience:

Supplementary Figures, Supplementary Note 1, Page S9

Supplementary Note 1: Microstructure of Ag NFs under tensile strain

(d) Contact angle of the Ag NFs film

Supplementary Figure 7 | Contact angle of Ag NFs on the PDMS substrate. (a) Contact angle of Ag NFs after transferring it on the PDMS substrate. (b) Contact angle of Ag NFs after fully drying. The Ag NFs will attach to the PDMS substrate owing to the Van der Waal's force after drying,

and the surface of the Ag NFs film will become hydrophobic when the Ag NFs are firmly bonded to the substrate. As shown in the Supplementary Figure 7, the contact angle of Ag NFs on the PDMS substrate via air gun blowing dry is 58.60°. After fully drying the Ag NFs at 100°C, it could be found that the contact angle becomes 127.73° which means the Ag NFs surface is hydrophobic, indicating that the water molecules in it have been completely evaporated and Ag NFs attach well to the PDMS substrate.

3. Several references can be added to help the readers to understand previous works related to this manuscript:

A. *Advanced Science* 5 (11), 1801146.

B. *Advanced Functional Materials* 27 (29), 1701138.

Response:

Thanks the reviewer for the suggestion. Several important previous reports about stretchable electrode have been added to the revised manuscript, as refs. 37, 38, 51 and 54.

REVIEWERS' COMMENTS:

Reviewer #1 (Remarks to the Author):

The authors have done a very careful job in responding to comments from the referees. I feel that the manuscript is suitable for publication in its current, revised form.

Reviewer #3 (Remarks to the Author):

This manuscript presents the silk-inspired highly stretchable transparent wireless Ag nanofiber (NF) coils. Regarding the characterization of stretchable coils for health care applications, the quality factor and radio frequency properties are optimized by considering effects of strain on NF networks. The authors definitely described important parameters such as geometrical characteristics, conductivity, permeability and adhesion of NF coils. In addition, the novelty of new Ag NF was demonstrated in detail by comparing the properties of Ag NFs with conventional Ag NW. I recommend this manuscript can be published in Nature Communications since the authors have prepared and included the reviewers' comments precisely.

Point to Point Response to the referees' reports
(comments in black, responses in blue):

Reviewer #1:

The authors have done a very careful job in responding to comments from the referees. I feel that the manuscript is suitable for publication in its current, revised form.

Answers:

We are delighted to read that Reviewer #1 approves the publication of our manuscript. Thank you very much.

Reviewer #3:

This manuscript presents the silk-inspired highly stretchable transparent wireless Ag nanofiber (NF) coils. Regarding the characterization of stretchable coils for health care applications, the quality factor and radio frequency properties are optimized by considering effects of strain on NF networks. The authors definitely described important parameters such as geometrical characteristics, conductivity, permeability and adhesion of NF coils. In addition, the novelty of new Ag NF was demonstrated in detail by comparing the properties of Ag NFs with conventional Ag NW. I recommend this manuscript can be published in Nature Communications since the authors have prepared and included the reviewers' comments precisely.

Answers:

We are delighted to read that Reviewer #3 approves the publication of our manuscript. Thank you very much.